# Human RNA Polymerase II Segregates from Genes and Nascent RNA and Transcribes in the Presence of DNA-Bound dCas9

**DOI:** 10.3390/ijms25158411

**Published:** 2024-08-01

**Authors:** João Pessoa, Célia Carvalho

**Affiliations:** 1Instituto de Medicina Molecular João Lobo Antunes, Faculdade de Medicina da Universidade de Lisboa, 1649-028 Lisboa, Portugal; celiacarv@edu.ulisboa.pt; 2Department of Medical Sciences and Institute of Biomedicine—iBiMED, University of Aveiro, 3810-193 Aveiro, Portugal

**Keywords:** gene array, RNA polymerase II, RNA transcription, nascent RNA, catalytically dead Cas9

## Abstract

RNA polymerase II (Pol II) dysfunction is frequently implied in human disease. Understanding its functional mechanism is essential for designing innovative therapeutic strategies. To visualize its supra-molecular interactions with genes and nascent RNA, we generated a human cell line carrying ~335 consecutive copies of a recombinant β-globin gene. Confocal microscopy showed that Pol II was not homogeneously concentrated around these identical gene copies. Moreover, Pol II signals partially overlapped with the genes and their nascent RNA, revealing extensive compartmentalization. Using a cell line carrying a single copy of the β-globin gene, we also tested if the binding of catalytically dead CRISPR-associated system 9 (dCas9) to different gene regions affected Pol II transcriptional activity. We assessed Pol II localization and nascent RNA levels using chromatin immunoprecipitation and droplet digital reverse transcription PCR, respectively. Some enrichment of transcriptionally paused Pol II accumulated in the promoter region was detected in a strand-specific way of gRNA binding, and there was no decrease in nascent RNA levels. Pol II preserved its transcriptional activity in the presence of DNA-bound dCas9. Our findings contribute further insight into the complex mechanism of mRNA transcription in human cells.

## 1. Introduction

In eukaryotic cells, RNA molecules are transcribed from DNA by three major RNA polymerases. RNA polymerase I transcribes most of the ribosomal RNAs. RNA polymerase II (Pol II) transcribes messenger RNAs (mRNAs) and most regulatory noncoding RNAs. RNA polymerase III transcribes transfer RNAs, the U6 small nuclear RNA, and the 5S ribosomal RNA [1].

Due to its critical role in mRNA biosynthesis, Pol II activity is a major determinant of protein homeostasis. Human malignancies, including cancer, autoimmunity, cardiovascular disease, and neurological disorders, are frequently associated with mutations in Pol II regulatory sequences and in proteins and RNAs that recognize such sequences [2], which are reflected in abnormal protein levels. For example, heart failure has been related to insufficient Pol II recruitment to metabolic gene promoters, possibly due to transcriptional coactivator dysregulation [3]. On the other hand, cancer frequently involves the up-regulation of transcription factors and other proteins [2]. As such, inhibition of the Pol II transcription machinery is a potential anticancer strategy [4]. Pol II may also be directly affected by diseases. Viral infections and DNA damage can modify the Pol II phosphorylation status and cause subunit depletion, both of which severely perturb Pol II function [5]. Therefore, understanding the functional mechanism of Pol II is valuable for designing innovative strategies for the therapeutic modulation of its transcriptional activity.

The functional mechanism of Pol II is remarkably complex, in part due to the unusually large number of molecular and supra-molecular interactions involved. Pol II is a protein complex containing 12 subunits [1]. Its activity requires additional and distinct proteins and protein complexes for the initiation, elongation, and termination of transcription [1]. Pol II function also depends on specific properties, including biomolecular condensation, which drives the reversible clustering of proximal Pol II complexes into nuclear compartments [6]. Pol II clustering increases transcription levels, which are usually proportional to cluster lifetime [7]. Nascent RNA also forms nuclear compartments seeded by noncoding RNA molecules at the transcription sites [8]. This mechanism promotes rapid local increases in RNA concentration and a competitive adaptation to environmental cues. Nevertheless, it remains poorly understood if, in single compartments, Pol II and nascent RNA are homogeneously distributed or physically segregated. Aberrations in biomolecular condensation processes have been related to human diseases [9]. As such, testing whether the Pol II and nascent RNA subcellular compartments are physically associated or segregated might uncover Pol II properties whose modulation could affect its transcriptional activity.

One of the strategies for Pol II transcriptional modulation is the binding of catalytically dead CRISPR-associated system 9 (dCas9) to a protein-coding gene [10,11]. In *Escherichia coli*, dCas9 binding to a gene could physically block mRNA transcription and decrease protein levels [10,11]. Although this effect is less efficient in eukaryotic cells [10], several promising applications have been developed. Multiple bindings of dCas9 to repetitive DNA sequences in human cells blocked the transcription of toxic RNAs related to microsatellite expansion diseases, including Huntington’s disease, myotonic dystrophy types 1 and 2, and others [12]. In Chinese hamster ovary cells, dCas9 binding to an essential gene increased the mRNA levels and the purification yield of a protein expressed from an adjacent recombinant gene [13]. In *Saccharomyces cerevisiae*, dCas9 binding to DNA modified chromatin condensation and generated novel sites of sense and antisense transcription [14]. Moreover, dCas9 binding could block not only DNA replication in *E. coli* and *S. cerevisiae* [15], but also the repair of DNA double-strand breaks in mouse embryonic stem cells [16]. These findings demonstrate that dCas9 binding to DNA can modulate not only gene expression but also DNA metabolism and repair, with promising applications in biomedicine and biotechnology. Notably, the use of dCas9 allows all these applications without modifying the genome.

dCas9 binding to DNA requires the hybridization of its attached guide RNA (gRNA) with one of the DNA strands. In both bacterial and human cells, dCas9-mediated transcriptional blocking was more effective when the gRNA hybridized with the coding strand of the gene [10,11,12,17]. Such a requirement might be exploited to strengthen unidirectional RNA transcription [17]. Nevertheless, the molecular basis of the decreased dCas9 blocking efficiency upon gRNA hybridization with the template strand remains poorly understood. In vitro protein assays demonstrated that DNA-bound dCas9 is a highly asymmetric roadblock [18]. Changing the gRNA-hybridized DNA strand rotates dCas9 180 degrees, possibly determining its fate in the event of colliding with Pol II. Consistently, gRNA hybridization with the template strand correlated with dCas9 removal by RNA polymerases from *E. coli* [18] and the T7 phage [16]. Nevertheless, it remains unclear if gRNA hybridization with the template strand induces the removal of dCas9 and/or Pol II from the gene in human cells.

Here, we studied the transcriptional mechanisms of human Pol II. To determine if the Pol II and nascent RNA compartments are physically associated or segregated, we visualized their supra-molecular arrangements from an array of short identical recombinant genes. To understand the decreased dCas9 transcriptional blocking upon gRNA hybridization with the template strand of a gene, we targeted dCas9 to different gene regions and monitored Pol II distribution throughout the gene body and nascent RNA levels.

## 2. Results

### 2.1. Recombinant Human Cell Lines to Study In Vivo Transcription

A complete human β-globin gene (*HBB*), from the initiation codon to the terminator sequence, was used as the starting point for the transgenic model construction. Besides a partial deletion in the second intron, this gene is fully functional. The post-poly(A) site region ensures proper transcription termination and 3′ end processing, as it occurs when the endogenous gene is transcribed [19,20]. Tandem repeats of ʎN-BoxB and MS2 stem-loops were inserted into introns 1 and 2, respectively, to enable the live visualization of the primary transcript as soon as it is produced at the site of transcription [21]. This construct was cloned under the control of a tetracycline-regulated CMV promoter, which permits the induction of high-level transcription by the addition of doxycycline.

Two model cell lines bearing a human β-globin transgene expressing a tetracycline-dependent regulatory protein were enrolled in the study. The U2OS-based system characteristically facilitates the insertion of multiple copies at the site of integration. In this system, after stable transfection with the recombinant construct, a monoclonal selection was performed. A cell clone was named “multiple-copy cell line”, in which a dCas9-GFP-expressing construct [22] was further integrated by stable transfection, followed by monoclonal selection (Figure 1A). This cell line can be transiently transfected with a gRNA-expressing plasmid to allow the binding of dCas9-GFP to a specific gene region. The Flip-In, HEK293T-based system allowed the insertion of one single copy of the transgene construct in the recombination site between two short flippase recognition target (FRT) sites. From this system, we produced a recombinant cell line hereafter named “single-copy cell line”, which was subsequently infected with lentiviral particles expressing dCas9-GFP, tet-On 3G transactivator, and a gRNA (from a mCherry-expressing vector) [23] recognizing telomeric repeats or a specific region within the human β-globin gene (Figure 1B). Several secondary cell lines, each one expressing a specific gRNA, were obtained by flow cytometry cell sorting.

We chose the human β-globin gene because of its relatively short size (~5.5 kbp, including both stem-loop arrays). Its short size keeps the entire length of each gene copy or single nascent RNA molecule below the resolution limit of optical microscopy. This requirement ensures that any structural features observed under the microscope are not intrinsic to single molecules, but a consequence of their supra-molecular interactions. 

To clarify the number of recombinant β-globin gene copies in the multiple-copy monoclonal cell line, we used a droplet digital PCR (ddPCR) assay and the single-copy transgenic cell line as a control. DNA purified from the cells was digested with a restriction enzyme that was confirmed to cut into the β-globin gene construct sequence but not in the PCR-amplified sequences or amplicons. With this approach, we intended to efficiently separate all the copies inserted into the genome. The digested DNA was mixed with all PCR reaction components, and then the mix was partitioned into thousands of oil emulsion droplets, generating an equal number of independent PCR reactions. Droplets with the target DNA sequence would be positive for PCR amplification, and droplets without it would be negative. After the readout of the number of PCR positive and negative droplets, a Poisson calculation is used to determine the counts of the DNA targets in the tested sample (Appendix A). Besides the property of absolute quantification, ddPCR results are not affected by the amplification efficiency of each primer pair, and its accurate quantification is linear throughout a wide dynamic range [24], making it the technique of choice to reliably compare between different amplification targets. Primer pairs specific to amplify a unique sequence in the BoxB array in intron 1 and another unique sequence in the MS2 array in intron 2 were used, respectively, to determine the number of copies of the transgene, in both the multiple-copy (Figure 1C) and single-copy (Figure 1D) cell lines. Furthermore, a primer pair targeting a region in the wild-type intron 2 of the β-globin gene that is absent in the β-globin construct was used to quantify the endogenous human gene as an internal control, which is known to be present with two copies per genome (Figure 1E). The three primer pairs were used for ddPCR in DNA samples from the single-copy and multiple-copy cell lines, as well as from a control cell line lacking recombinant β-globin (Figure 1F). The target copy number concentration was confirmed to vary linearly over a wide range of serial DNA dilutions (Appendix A) and the accuracy of the technique was confirmed by the determination of a single integration in the Flip-In single-copy cell line (1.15 ± 0.17 copies per cell). Finally, the number of recombinant β-globin gene copies in the multiple-copy integration cell clone was calculated to be 335.6 ± 15.3 copies per cell (Figure 1G).

### 2.2. Human β-globin Nascent RNA Segregates from Its Originating Genes

Using the U2OS cell line carrying an array of recombinant β-globin gene copies (Figure 1C), we first addressed how the gene array interacts with its nascent RNA by visualizing these two elements. For nascent RNA visualization, we used two standard methodologies. One was the fluorescent labeling of the 24 MS2 stem loops located in intron 2 (Figure 2A). The other approach was RNA fluorescence in situ hybridization (FISH) using a probe recognizing exon 2 (Figure 3A). A major difference between the two methodologies is the recruitment of a large protein mass to nascent RNA molecules upon stem-loop labeling in living cells. We have previously shown that the full array of 24 MS2 stem-loops binds an average of 30 units of a fusion protein containing one MS2 coat protein (MCP) and one GFP molecule [21]. Considering a molecular weight of 26.9 kDa for GFP and of 13.7 kDa for MCP, an average of 1.22 MDa of protein mass will be recruited to each single mRNA molecule, likely affecting the nascent RNA interactions with other molecules.

U2OS cells were co-transfected with plasmids coding for a gRNA and an MCP-mCherry fusion protein (to target stably expressed dCas9-GFP to the exon 3 of the gene copies and fluorescently label MS2-tagged nascent RNA, respectively) and treated with 1 µg/mL doxycycline to ensure high levels of RNA transcription from each gene copy in the array (Figure 2A). Living cells were visualized under a confocal microscope. We observed the dCas9-GFP-labeled gene copies as a globular shape in the green emission channel. Their nascent RNA was visualized in the red emission channel. Three-dimensional (3D) reconstructions of confocal micrographs where both signals were visible revealed a minor overlap, a major overlap, or a full overlap between the gene array and its MS2-labeled nascent RNA (Figure 2B and Appendix A). A major overlap was the most frequent observation (Figure 2C). In each 3D reconstruction, we measured the longest diameters of the gene array and MS2-labeled nascent RNA signals. We observed that the nascent RNA signal was smaller than that of the gene array (Figure 2D), indicating that RNA transcription was not homogeneous throughout the gene array. We also quantified the intensities of the two fluorescent signals (Appendix A). Furthermore, we observed a positive correlation between the diameters of both signals (Figure 2E) as well as between their intensities (Appendix A), with no evident association with their overlap extent. The variable dimension of the gene array signal suggests variations in its chromatin condensation state. A less condensed gene array might spatially expand its transcription regions and enhance transcription levels, possibly contributing to a direct correlation between the signal dimensions of the gene array and its nascent RNA. Furthermore, the positive correlation between the intensities of both signals suggests that enhanced chromatin decondensation was associated with both its labeling with higher levels of dCas9-GFP and the transcription of larger amounts of RNA.

To assess the basal levels of RNA transcription in our transgene system, we performed this experiment without adding doxycycline. In a preliminary assay, we visualized both the gene array and its MS2-labeled nascent RNA, having also performed 3D reconstructions. We could easily visualize both minimal and extensive RNA transcription in the absence of doxycycline-mediated transcription activation (Appendix A). When abundant amounts of nascent RNA were visualized, a major overlap between the gene array and the nascent RNA signals was also the most frequent observation (Appendix A). The dimension of the nascent RNA signal was significantly smaller than that of the gene array (Appendix A). The intensities of the nascent RNA signals were, in most cases, relatively low (Appendix A). We assessed the correlations between the dimensions of both signals (Appendix A) and also between their intensities (Appendix A). In both cases, there was a poor correlation. This lack of positive correlation suggests that, in the absence of transcription induction, the levels of nascent RNA are stochastically variable. They are not necessarily increased during enhanced chromatin decondensation, which likely favors both increased gene array signal length and its enhanced labeling with dCas9-GFP. Overall, this preliminary assay indicated that our gene system is able to transcribe RNA in the absence of transcription induction.

We repeated this assay, detecting nascent RNA by FISH (Figure 3A). As in above, cells were transfected with a gRNA-coding plasmid that targeted dCas9-GFP to exon 3 of each gene copy in the array and treated with doxycycline. Cells were fixed, and nascent RNA was subsequently labeled by FISH. Under the confocal microscope, we observed the gene array and its FISH-labeled nascent RNA. 3D reconstructions where both signals were visible mostly revealed a minor overlap, major overlap, or full overlap between the two elements (Figure 3B), although one minimal overlap case was recorded (Appendix A). In the previous assay, a major overlap between both signals was the most frequent observation (Figure 3C). In each 3D reconstruction, we measured the largest diameter of the gene array and FISH-labeled nascent RNA signals. We observed that the nascent RNA signal was generally smaller than that of the gene array (Figure 3D), indicating that RNA transcription remained heterogenous throughout the gene array. We also quantified the intensities of both signals (Appendix A), which revealed their heterogeneity. There was no association between signal diameter and overlap extent; however, the positive correlation between the diameters and intensities of both signals was lost under FISH labeling of the nascent RNA (Figure 3E and Appendix A, respectively). Such correlation loss might be due to the less sensitive detection limit of the FISH probe used relative to MS2-labeling. Each recombinant β-globin nascent RNA molecule recruits an average of 30 MCP-mCherry molecules [21]; however, it contains only one binding site for the FISH probe. As such, we hypothesize that the visualization of FISH-labeled nascent RNA requires more RNA condensation than MS2-labeling. Therefore, less condensed nascent RNA signals might not be readily detected by FISH, disrupting the positive correlation between the diameters and intensities of the gene array and FISH-labeled nascent RNA signals.

### 2.3. mRNA Transcription Is Heterogenous among Identical Human β-globin Genes

Following the above observations, we hypothesized that the generally observed major overlap between the gene arrays and their nascent RNA could be a consequence of the interaction between the gene array and transcribing Pol II. To test this hypothesis, we visualized both the gene array and Pol II. Cells were transfected with a gRNA-expressing plasmid for targeting stably expressed dCas9-GFP to the exon 3 of each of the ~335 β-globin gene copies and treated with 1 µg/mL doxycycline to ensure high levels of Pol II recruitment to each gene copy in the array. Cells were subsequently fixed, and Pol II was visualized by immunofluorescence using an antibody recognizing its C-terminal domain phosphorylated at serine 2 (Figure 4A), a phosphorylation state related to transcription elongation [25].

Under the confocal microscope, we observed the gene array and Pol II signals. Pol II was spread throughout the cell nucleus, although it was depleted from the nucleoli (Figure 4B). Z-stack micrographs were captured from cells containing both signals. 3D reconstructions of the z-stacks revealed the localization of Pol II within the gene array. There were two interaction forms observed: a more frequent major overlap and a less frequent full overlap between Pol II and the gene array (Figure 4C and Appendix A). Cases where Pol II was found completely outside of the gene array (with minimal overlap) were not found. In each 3D reconstruction, we measured the longest diameter of the gene array and the Pol II signals. We observed that the Pol II signal was smaller than the gene array signal (Figure 4D), indicating that the Pol II was concentrated only in a limited region of the gene array. The intensities of the Pol II signals were highly heterogeneous relative to those of the gene array signals (Appendix A), indicating that the gene array was able to recruit heterogeneous amounts of Pol II. We also observed a positive correlation between the diameters (Figure 4E) and intensities (Appendix A) of both signals, with no evident association with their overlap extent. The variations in the diameter of the gene array signal could be due to the variable extents of chromatin condensation. When the gene array was less condensed, it might be able to accommodate higher levels of Pol II. Furthermore, the lower correlation between signal intensities suggests random fluctuations in the levels of Pol II recruited to the gene array. Our observation indicates that, although all gene copies in the array were identical, their Pol II distribution and levels were not homogeneous. The two interaction forms observed suggest that the Pol II signal travels throughout the gene array, sequentially transcribing different gene copies.

### 2.4. Human β-globin Nascent RNA Segregates from RNA Polymerase II

From the above observations, we hypothesized that the compacted morphology of the Pol II signal might affect its interactions with its nascent RNA. We tested this hypothesis by visualizing both elements. We combined Pol II immunofluorescence with MS2-labeling of the nascent RNA (Figure 5A). Cells were transfected with an MCP-GFP-expressing plasmid, treated with doxycycline, fixed, and subjected to immunofluorescence, as in the previous section. Under the microscope, we visualized Pol II (red channel) and MCP-GFP (green channel) diffused throughout the nucleus and also concentrated at transcription sites, with higher intensity at the transgene array locus. After selecting cells where both signals were visible, we did 3D reconstructions of their interaction and found two main class distributions (Figure 5B). There was mostly a minor overlap between Pol II and nascent RNA or a major overlap between both (Figure 5C and Appendix A). In each 3D reconstruction, we measured the largest diameter of the Pol II and nascent RNA signals. We observed that their average diameters were similar (Figure 5D), suggesting that, on average, nascent RNA extensively interacted with the Pol II signal. The intensities of both signals were also assessed (Appendix A). We also observed a positive correlation between the diameters of both signals, with no evident association with their overlap extent (Figure 5E). Such a positive correlation suggests extensive physical interactions between the two signals, in which an increase in the size of the Pol II signal goes along with and favors an increase in the size of the primary transcript (MS2) signal. Nevertheless, the correlation between the intensities of both signals was not so apparent (Appendix A), meaning that higher levels of Pol II are not accompanied by the accumulation of unspliced (MS2) primary transcript at the transcription site, in accordance with a lack of interference with co-transcriptional splicing.

Then, we repeated the Pol II and nascent RNA double labeling but replaced MS2 stem-loop labeling with RNA FISH (Figure 6A). Under the microscope, we visualized Pol II in the green emission channel and the nascent RNA exon in the red emission channel. As in above, we performed 3D reconstructions (Figure 6B). We found minimal overlap and minor overlap classes; however, the major overlap was the most frequent observation (Figure 6C and Appendix A). In each 3D reconstruction, we measured the largest diameter of the Pol II and nascent RNA signals. As in above, we observed that their average signal diameters were similar (Figure 6D), also suggesting an extensive association between Pol II and nascent RNA. Furthermore, the signal intensities of Pol II seemed to be more variable than those of nascent RNA (Appendix A). However, the positive correlation between the diameters of both signals was less evident for FISH-labeled nascent RNA (Figure 6E) relative to MS2-labeled nascent RNA (Figure 5E). As observed in Figure 3E, this difference could be due to the less sensitive detection limit of the FISH probe relative to MS2-labeling. A poor correlation was also observed between the intensities of both signals (Appendix A), suggesting that the levels of both Pol II and its associated FISH-labeled nascent RNA were not directly proportional. 

Regardless of the nascent RNA labeling method, there is likely a dynamic equilibrium among the multiple types of overlap between Pol II and its nascent RNA. The average major overlap between both could result from a rapid and extensive interaction of nascent RNA with its processing proteins, resulting in its partial dissociation from Pol II.

### 2.5. dCas9 Binding to a Human β-globin Gene Is Stable over Time

The second aim of the present study was to test if dCas9 binding to a Pol II-transcribed gene (with the gRNA hybridized with the template strand) causes the removal of dCas9 and/or Pol II from the gene. First, we tested for any potential dCas9 removal. We assessed if, under Pol II transcription, dCas9 binding to the gene array is transient (with binding, unbinding, and re-binding cycles) or stable over time by fluorescence recovery after photobleaching (FRAP). In a FRAP experiment, a fluorescent probe bound to a target of interest is photobleached, and signal recovery is monitored over time. As photobleaching is irreversible, signal recovery depends on the de novo binding of the probe.

We targeted dCas9-GFP to exon 2 of the gene array using a gRNA that hybridizes with the template strand of the gene (Figure 7A). The signal was photobleached in the absence of doxycycline, and its recovery over time was visually assessed. Fluorescence intensity was quantified in a region of interest within single cells before (normalized to the unit) and after (normalized to zero) photobleaching of dCas9-GFP. We did not detect any signal recovery for almost 10 min after photobleaching (Appendix A and Figure 7B). Then, we repeated the assay in the presence of 1 µg/mL doxycycline to induce high transcription levels. Signal recovery was monitored over time, and, as observed in the absence of doxycycline, signal recovery was undetected for almost 10 min after photobleaching (Appendix A and Figure 7C). Overall, FRAP assays have shown undetectable de novo binding of dCas9 to exon 2 of the recombinant gene. The absence of signal recovery suggests that dCas9-GFP binding to exon 2 of human β-globin is stable over time, both at basal and high transcription levels. Consequently, dCas9 was likely not removed from the human β-globin gene by transcribing Pol II. We also tested if dCas9 prevents Pol II recruitment to the recombinant gene promoter by targeting it to the seven operator repeats (Figure 7D) in the presence of doxycycline. Normalized signal recovery revealed a slow recovery of the signal (Appendix A and Figure 7E), probably due to a partial dCas9 displacement from the operator through binding competition with the tetracycline transactivator protein (rtTA). As a control experiment, we overexpressed dCas9-GFP in these cells, which results in its nonspecific binding to nucleoli. A region of interest was defined within a nucleolus, and nonspecifically bound dCas9-GFP was photobleached. Quantified fluorescence intensity revealed signal recovery in about 20 s (Appendix A and Figure 7F). This finding confirms that dCas9-GFP binding to nucleoli is temporally unstable, resulting in high turnover rates and, consequently, rapid recovery of fluorescence after photobleaching. Overall, we observed that, under high levels of Pol II transcription, the binding of dCas9-GFP to exon 2 of the β-globin gene is stable over time (Appendix A and Figure 7C).

### 2.6. dCas9 Has Little Effect on the Levels of Paused RNA Polymerase II in the Promoter Region

We then tested if dCas9 binding to the gene affects Pol II transcriptional activity. For this goal, we combined two methodologies: chromatin immunoprecipitation (ChIP) and droplet digital reverse transcription PCR (ddRT-PCR). ChIP was used to test if dCas9 binding increases the accumulation of transcriptionally paused Pol II in the promoter region of the β-globin gene body. ddRT-PCR was used to quantify any alterations in nascent RNA levels. We conducted these experiments in the cell line containing only a single copy of the recombinant gene (Figure 1B), also expressed from a doxycycline-inducible promoter [21]. Using this cell line and a lentiviral system [23], we stably integrated constructs expressing dCas9-GFP and a gRNA-binding telomeric repeat (negative control for binding to the β-globin gene) or one of the three β-globin gene exons. All three gRNAs hybridized with the template strand of the gene. The binding of dCas9-GFP to telomeric repeat DNA generates an array of diffraction-limited spots in the cell nucleus [23], which we have reproduced in U2OS cells (Figure 8A, inset). About 24 h after 1 µg/mL doxycycline treatments, dCas9-GFP and Pol II distributions throughout the gene body were monitored by ChIP, and nascent RNA levels were quantified by ddRT-PCR.

In ChIP assays, chromatin was cross-linked, fragmented into ~200 bp pieces (Appendix A), and immunoprecipitated with antibodies recognizing dCas9-bound GFP and/or the Pol II C-terminal domain phosphorylated at serine 5, a phosphorylation state related to transcriptional pausing [25]. Immunoprecipitated DNA fragments were amplified by quantitative PCR (qPCR) using primer pairs binding to the promoter/exon 1, exon 2, or exon 3 of the recombinant gene or to an intergenic region (negative control). Relative DNA levels associated with either dCas9 or Pol II were compared. 

As expected, ChIP results showed undetectable dCas9 enrichment in any of the amplified β-globin gene regions upon dCas9 binding to telomeric DNA (Figure 8A). Pol II was mostly distributed in the promoter and exon 1 regions (Figure 8B). Upon dCas9 binding to one of the recombinant gene exons, ChIP reveals enrichment in dCas9 in the first (Figure 8C), second (Figure 8E), and third (Figure 8G) exons. The levels of transcriptionally paused Pol II in the promoter/exon 1 region were identical among the control and the three conditions (Figure 8B,D,F,H). We further hypothesized that dCas9 might block transcriptional activity in a strand-specific way through gRNA-DNA hybridization, in agreement with previous findings [10,11,12]. To test this hypothesis, we moved the gRNA to the coding strand of exon 1. ChIP confirmed dCas9 binding to exon 1 (Figure 8I). Interestingly, there was an increase in paused Pol II levels in the promoter/exon 1 region when the gRNA was hybridized with the coding strand of exon 1 (Figure 8J). These results are suggestive of partial transcriptional blocking by dCas9 in a strand-specific way of gRNA hybridization.

### 2.7. RNA Polymerase II Transcribes in the Presence of DNA-Bound dCas9

We also assessed if dCas9 binding to the recombinant β-globin gene affected its nascent RNA levels by one-step ddRT-PCR (Appendix A). Nascent RNA was isolated by chromatin fractionation of the above-described conditions and added to a ddRT-PCR reaction mix (containing a reverse transcriptase). The reaction mixture also contained primer pairs for amplification of the complementary DNA (cDNA) from β-globin and glyceraldehyde 3-phosphate dehydrogenase (GAPDH) at the intron 1–exon 2 and the intron 2–exon 3 junctions, respectively. GAPDH served as an internal control for β-globin normalization. Using ddRT-PCR, we did absolute quantifications of the cDNA levels of human β-globin and GAPDH (Figure 9A), with the latter being used for β-globin normalization (Figure 9B). The human β-globin nascent RNA levels were essentially identical whether dCas9 was targeted to telomeric repeats or β-globin exon 1 or exon 2. Interestingly, there was a statistically significant ~50% increase in β-globin nascent RNA levels when dCas9 was targeted to exon 3 (Figure 9B, top). This nascent RNA accumulation is compatible with a partial blocking of 3′ end processing and release of the transcriptional site (Figure 9B, down). As a negative control for recombinant β-globin transcription, we also tested an identical cell line without the recombinant β-globin gene, where we could detect only minimum nascent RNA levels (Figure 9B, top). Taken together, our ddRT-PCR results show that the levels of mRNA transcription were not decreased upon dCas9 binding to the β-globin gene. Nevertheless, they suggest a potential accumulation of nascent RNA in one of the tested conditions.

We repeated this assay using the cell line in which dCas9 is bound to exon 1, with the gRNA hybridized with the coding strand of the recombinant gene. Nascent RNA levels were also quantified for the β-globin gene and the internal control (GAPDH) using ddRT-PCR (Figure 9C). GAPDH-normalized data indicated identical levels of β-globin nascent RNA in cells where the gRNA was bound to telomeric DNA or to the coding strand of β-globin gene exon 1 (Figure 9D, top), which is compatible with the lack of transcriptional blocking (Figure 9D, down). Altogether, these results show that dCas9 binding to a recombinant β-globin gene did not affect mRNA transcription, even when its gRNA was hybridized with the coding strand of the gene.

## 3. Discussion

In the present research, we studied two features of human Pol II. One was the extensive compartmentalization of its supra-molecular interactions with genes and nascent RNA. This property could be required for efficient mRNA transcription and may be affected by human disease. The second was the Pol II capacity to maintain transcriptional activity in the presence of DNA-bound dCas9. This capacity demonstrates not only the versatility of Pol II in overcoming a physical obstacle but also the limitation of dCas9 as a tool for inducing transcriptional interference. The potential implications of these findings are discussed below.

We visualized the supra-molecular interactions among an array of ~335 recombinant human β-globin genes, Pol II, and their nascent RNA molecules. We quantified both the longest diameters and intensities of fluorescent signals to search for correlations between signal pairs. While the measurement of the signal diameter is more reproducible among independent microscopy experiments, signal intensity provides a more direct readout of the number of fluorescent molecules bound to the target. Although all genes in the array were identical, they were not uniformly transcribed (Figure 2B, Figure 3B and Appendix A). Furthermore, Pol II did not simultaneously overlap with all of them (Figure 4B and Appendix A). The confined localization of Pol II within the gene array implies that Pol II is not equally transcribing all the genes simultaneously. In our work, nascent RNA also formed compartments with extensive overlap with Pol II (Figure 5B, Figure 6B and Appendix A). The extensive segregation between Pol II and nascent RNA is striking, considering the relatively short length of each nascent RNA molecule (about 3.7 kbp) and the previously characterized interactions between nascent RNA and the C-terminal domain of Pol II [26,27]. Such extensive segregation might result from the high affinity between nascent RNA and the molecules responsible for its processing. Super-resolution microscopy studies also identified clusters of transcribing Pol II and partially overlapping nanoscale domains of nascent RNA [28]. However, the labeling method used was not specific for short mRNA molecules, as in our work. In our system, as a likely consequence of the segregation between Pol II and nascent RNA, nascent RNA also exhibited a partial overlap with the gene array (Figure 2B, Figure 3B, and Appendix A). Consistently, super-resolution microscopy studies also visualized segregation between nascent RNA and nucleosomes [28]. 

Similarly to Pol II, RNA polymerase I has also been shown to cluster at nucleolar transcription sites [29]. Nucleoli exhibit DNA–RNA segregation as a consequence of several partial gene silencing mechanisms, including DNA methylation [30], histone modifications [31], chromatin remodeling [32,33], and interactions with intergenic RNAs [34]. Nevertheless, these mechanisms are likely absent from our gene array. We propose that the biomolecular condensation properties of Pol II and nascent RNA could promote DNA–RNA segregation in the absence of partial gene silencing mechanisms. 

The formation of chromatin loops during RNA transcription has been extensively documented [35]. These loops are not exclusive to long DNA regions; they are also found in transcribed sequences as short as 1 kbp [36]. Each gene copy in our array is about 5.5 kbp long, compatible with the formation of chromatin loops. Transcription from chromatin loops was also observed in RNA polymerase I-transcribed ribosomal RNA in human and mouse cells [37]. In our gene array, Pol II was likely able to cluster through its biomolecular condensation properties. Pol II clusters could have been formed at chromatin loops, further promoting their pulling out from the gene array to facilitate mRNA transcription (Figure 10A). The formation of chromatin loops in central regions of the gene array could have resulted in fully overlapped signals between the gene array and Pol II. On the other hand, chromatin loops might have been pulled out from peripheral regions of the gene array, resulting in partially overlapped signals (Figure 10A). In chromatin loops, Pol II clusters likely promote the physical separation, not only between Pol II and the gene array but also between Pol II and its nascent RNA (Figure 10B). This is supported by transcriptional activity affecting chromatin mobility independently of its condensation state [38]. The decreased overlap of Pol II with genes and their nascent RNAs could be a feature of human mRNA transcription. Alterations in the segregation extent between Pol II and genes and/or between Pol II and nascent RNA might affect the rates and fidelity of transcription, with possible implications for human disease.

We have also addressed the transcriptional consequences of dCas9 binding to DNA. The binding of dCas9-GFP to exon 2 of a recombinant β-globin gene (using a gRNA hybridizing the template strand of the gene) was stable over time, both in the absence (Figure 7B) and presence (Figure 7C) of high Pol II transcription levels. Minor or negligible recovery of FRAP signals has also been observed in other studies, including dCas9-GFP binding to mouse B2-type short interspersed nuclear elements [39] and telomeric repeats [40,41]. These observations further support long-lasting binding to DNA as a general property of dCas9, with its average residence time between three and four hours [42]. Stable binding of dCas9 to a gene indicates that it was likely not removed by transcribing Pol II. Moreover, this property potentially enables dCas9 to function as an effective roadblock to Pol II. This hypothesis was tested in a cell line carrying a single copy of a recombinant β-globin gene by ChIP and ddRT-PCR. In comparison with telomeric DNA binding (Figure 8A), when dCas9 was bound to the first, second, or third exons of the recombinant β-globin gene (Figure 8C,E,G), there was no significant accumulation of transcriptionally paused Pol II in the promoter/exon 1 region of the gene (Figure 8D,F,H). In addition, the levels of nascent RNA were increased only upon binding of dCas9-GFP to exon 3 (Figure 9B). When the gRNA was switched to the coding strand of exon 1 (Figure 8I), the accumulation of transcriptionally paused Pol II increased (Figure 8J), but the levels of nascent RNA (Figure 9D) were not significantly affected. Targeting the gRNA to the coding strand has been correlated with higher efficiency of transcriptional interference in both bacterial and human cells [10,11,12]. Nevertheless, the efficiency of Cas9/dCas9 biotechnological applications usually depends on the gRNA binding sequence within a specific DNA region. This observation might explain the transcriptional interference in our gene system upon gRNA hybridization with the coding strand. Importantly, multiple bindings of dCas9 to repetitive DNA sequences were sufficient to robustly block mRNA transcription in human cells [12]. In our recombinant gene, the binding of a single dCas9 molecule was not sufficient for the same purpose, indicating that multiple bindings are critical. Furthermore, these findings uncover a limitation of using dCas9 to block transcription of toxic RNA. Our observations indicate that Pol II is able to maintain its transcriptional activity in the presence of DNA-bound dCas9. Here, we assessed the impact of dCas9 binding on Pol II distribution and nascent RNA levels, providing direct measurements of the response at the transcriptional level. We conclude that, in our recombinant gene, dCas9 was unable to block a high flow of transcribing Pol II (induced with 1 µg/mL doxycycline) while remaining bound to its target, with a negligible impact on transcription (Figure 10C). This observation is consistent with the Pol II capacity for transcribing through nucleosomes with minor histone displacement [43], in which FRAP has been used to monitor the binding dynamics between DNA and histones [44].

In addition to the above-described findings, our work also introduces a cell system for studying transcriptional architecture. Our cell line, which contains an array of ~335 identical β-globin gene copies, enabled the fluorescent imaging of a portion of chromatin composed of short identical recombinant genes. Here, we have only visualized its associated Pol II and nascent RNA. Nevertheless, additional factors could be fluorescently labeled and visualized in future studies, including transcription factors, splicing factors, noncoding RNAs, and other molecules associated with mRNA transcription. Moreover, this cell line enabled visualizing the relative positioning of DNA, Pol II, and nascent RNA without the need for super-resolution microscopy. Notably, the simultaneous visualization of the gene array and its MS2-labeled nascent RNA was achieved in living cells. When live-cell visualization is attainable, dynamic processes can be tracked over time. However, a preliminary experiment (Appendix A) has shown that this cell line reveals detectable mRNA transcription signals in the absence of transcription induction with doxycycline. This significant basal transcription, which contrasts with early observations in a distinct doxycycline-inducible system [45], limits kinetic studies and the feasibility of visualizing the supra-molecular structures associated with full transcription silencing or shorty after transcription induction, such as differences in signal overlap. Furthermore, the recombinant gene integration sites have not, so far, been experimentally determined in single-copy or multiple-copy cell lines. Additional limitations of our work include the limited spatial resolution of ChIP, which might have contributed to masking minor Pol II blocking events (Figure 8F). Future steps could include the visualization of the total Pol II levels (including its unphosphorylated and serine 5-phosphorylated forms) by immunofluorescence in the multiple-copy cell line, which could provide a more complete depiction of Pol II interactions with the gene array. Concerning the potential blocking of transcriptional elongation by dCas9, a possible future test could be the simultaneous targeting of the three gene exons by dCas9. This approach should create a more robust barrier to Pol II transcriptional elongation and contribute to better understanding the absence of Pol II blocking observed herein. In addition, it could be useful to address, by sequencing approaches, if nascent RNA was fragmented in the presence of dCas9 binding to the β-globin gene. 

In conclusion, we have shown extensive compartmentalization among an array of short identical recombinant genes, Pol II, and nascent RNA. Moreover, Pol II could retain its transcriptional activity in the presence of DNA-bound dCas9. These findings contribute insight into the mechanism of RNA transcription by Pol II, which could be useful for designing therapeutic modulation strategies. These strategies could aim at changing Pol II transcription levels by affecting its compartmentalization or by effectively blocking its activity.

## 4. Materials and Methods

### 4.1. Cell Lines

All cell lines in this study were cultured in Dulbecco’s Modified Eagle Medium (DMEM; Gibco, New York, NY, USA) supplemented with 10% fetal bovine serum (Gibco, New York, NY, USA). The cell line carrying a single recombinant human β-globin gene copy was previously generated [21]. The cell line carrying multiple copies of a recombinant human β-globin gene was generated in this study using genetically modified U2OS cells (Clontech, Mountain View, CA, USA), as described below. Cell lines were authenticated through the determination of the respective recombinant human β-globin gene copy number.

### 4.2. Cloning of gRNA-Expressing Sequences

For gRNA expression, the PX459 plasmid (Addgene plasmid #62988) was modified with an optimized gRNA design [23]. Its wild-type Cas9-coding gene was removed by restriction digestion, gel extraction, and religation of the Cas9-free plasmid. For cloning of each gRNA-expressing construct, the modified PX459 plasmid was digested with BbsI (New England Biolabs, Ipswich, MA, USA), dephosphorylated, and gel-purified. DNA oligonucleotides (Appendix A) were phosphorylated, annealed, and ligated into the digested plasmid using T4 DNA ligase (Thermo Fisher Scientific, Waltham, MA, USA). Ligation products were transformed into competent *E. coli* DH5α cells and purified using the NZYtech plasmid miniprep kit (NZYtech, Lisbon, Portugal). Plasmids were eluted in 10 mM Tris-HCl pH 8.0 prepared in water deionized in a Milli-Q system (Millipore Corporation, Billerica, MA, USA). Insert ligation was confirmed by Sanger sequencing.

For stable expression of dCas9-GFP and a gRNA in cell lines containing a single recombinant human β-globin gene copy, gRNA sequences recognizing the gene were re-cloned into a lentiviral gRNA vector with optimized design [23]. gRNAs were PCR-amplified from the modified PX459 vector with the primers listed in Appendix A. Phusion DNA Polymerase (Thermo Fisher Scientific, Waltham, MA, USA) was used, according to the enzyme’s manual. PCR products and the lentiviral gRNA-expressing plasmid were digested with BstXI and XhoI and purified. Digested PCR products were ligated into the lentiviral gRNA vector (which originally contained a telomeric repeat recognition sequence) using T4 DNA ligase (Thermo Fisher Scientific, Waltham, MA, USA). Insert ligation was confirmed by Sanger sequencing.

### 4.3. Generation of a Human Cell Line Carrying Multiple Copies of a Recombinant Human β-globin Gene

The human β-globin genomic construct used consists of the human gene from cytogenetic band 11p15.4, carrying a deletion of 593 bp on intron 2–3 between RsaI and SspI restriction sites. The construct was previously modified on intron 1 (at the BpiI restriction site) by insertion of an array of 25 ʎN-BoxB binding sites and on intron 2 (at the Psp5II restriction site) by insertion of an array of 24 MS2-coat binding sites. This construct was previously detailed [21]. A blunted NcoI-Acc65I insert containing the genomic construct, from initiation codon to approximately 1800 bp past the poly(A) site (at B*gl*II restriction site), was recombined with the blunted Acc65I site of the pTRE-tight vector (Clontech, Mountain View, CA, USA).

The construct described above was transfected into Tet-On U2OS-derived cells (Clontech; Mountain View, CA, USA), which provides a regulated, high-level gene expression system [46] upon doxycycline induction. These cells have a transgenic insertion of the plasmid pTet-On, which confers resistance to the G418 antibiotic (or neomycin). This gene drives the expression of the rtTA regulatory protein. The construct was linearized by digestion with the PvuI restriction enzyme and further co-transfected with pTK. Hyg plasmid linearized with VspI in a 9:1 proportion. The recombinant cells were selected by culture on growth medium containing 0.6 mg/mL hygromycin (Roche, Basel, Switzerland). Clones were screened by expression of the β-globin construct after induction with 6 mg/mL doxycycline (Clontech, Mountain View, CA, USA) by RT-PCR. 

The pCAG-dCas9-GFP plasmid [22] was further transfected using the Neon electroporation kit (Life Technologies, Waltham, MA, USA). 4 × 10^5^ cells were resuspended in 120 µL Neon buffer R, and 5 µg pCAG-dCas9-GFP plasmid was added. Cells were electroporated by applying three 10 ms pulses of 1400 V through a 100 µL Neon tip. Electroporated cells were seeded onto a 15 cm culture plate, and 10 µg/mL blasticidin S (Thermo Fisher Scientific, Waltham, MA, USA) were added after 24 h for the selection of positive clones.

### 4.4. Stable Integration of dCas9-GFP and gRNA into a Human Cell Line Carrying a Single Copy of a Recombinant Human β-globin Gene

Cells stably expressing dCas9-GFP and a gRNA were generated by lentiviral infection of the cell line carrying a single copy of the recombinant human β-globin gene [21]. Lentiviral suspensions used for infection expressed dCas9-GFP, the Tet-ON-3G transactivation factor, and a gRNA recognizing telomeric repeats or either human β-globin exon 1, exon 2, or exon 3 in the template strand or exon 1 in the coding strand (Appendix A). The lentiviral gRNA vector also expresses mCherry.

To produce each lentiviral suspension, HEK293T cells were grown in a T75 flask (Thermo Fisher Scientific, Waltham, MA, USA) to 50–70% confluency and transfected with 1 µg pMD2.G, 8 µg pCMV-dR8.91, and 9 µg transfer vector (either dCas9-GFP, Tet-ON-3G, or a lentiviral gRNA plasmid) using Lipofectamine 2000 (Invitrogen, Waltham, MA, USA). On day 2, medium was replaced (12 mL/T75 flask). On day 3, 12 mL lentiviral suspension were collected from each flask and stored at 4 ºC; 12 mL fresh medium were added to each flask. On day 4, a second portion of 12 mL lentiviral suspension was collected from each flask, pooled with the corresponding suspension collected on day 3, filtered through a 0.45 µm filter, and concentrated by ultracentrifugation. Ultracentrifugation was performed at 25,000 rpm for 2 h at 4 °C in a Beckman SW41 rotor. The supernatant was drained, and lentiviral pellets were resuspended in fresh medium by pipetting. Each 24 mL portion of centrifuged lentiviral suspension was resuspended in 2 mL fresh medium.

For the infection of the cell line carrying a single recombinant human β-globin gene copy, cells were seeded on 3.5-cm plates and grown to 50–70% confluency. They were infected with 1 mL dCas9-GFP, 1 mL Tet-ON-3G, and 2 mL gRNA lentiviral suspensions, in the presence of 5 µg/mL polybrene (Sigma-Aldrich, St. Louis, MO, USA). On day 5, 2 µg/mL puromycin (Sigma-Aldrich, St. Louis, MO, USA) was added to select for transduced cells. The antibiotic was kept for 5–7 days, and on day 8 after lentiviral infection, cells exhibiting simultaneous GFP and mCherry fluorescence were sorted by flow cytometry at the Flow Cytometry Facility of Instituto de Medicina Molecular João Lobo Antunes.

### 4.5. Recombinant β-globin Copy Number Determination by ddPCR

For recombinant β-globin gene copy number determination, genomic DNA was isolated and analyzed by ddPCR. Cells were harvested and resuspended in phosphate-buffered saline (PBS) buffer pH 7.4. Histones were degraded by overnight incubation with 200 µg/mL Proteinase K (Thermo Fisher Scientific, Waltham, MA, USA). Genomic DNA was extracted in a phenol/chloroform/isoamyl alcohol solution (Thermo Fisher Scientific, Waltham, MA, USA), precipitated in ammonium acetate and ethanol, and dissolved in deionized water. Isolated genomic DNA was analyzed by ddPCR assay using the QX200™ Droplet Digital™ PCR System (Bio-Rad, Hercules, CA, USA) and EvaGreen Digital PCR Supermix according to the manufacturer’s instructions. Genomic DNA was analyzed with primer sets that were specific to the stem-loops in the recombinant gene as well as to the endogenous human β-globin only (taking advantage of a partial deletion in the second intron of the recombinant gene, between *Rsa*I and *Ssp*I restriction sites). As a negative control, genomic DNA from plain U2OS cells was also analyzed. Primers are listed in Appendix A.

Purified genomic DNA was digested overnight with the HaeIII restriction enzyme (New England BioLabs, Ipswitch, MA, USA) before concentration determination in NanoDrop 2000 UV-Vis spectrometer (Thermo Fisher Scientific, Waltham, MA, USA). For ddPCR, 20 μL reaction mixtures were prepared, containing 2 ng of template DNA, 0.5 μL of HaeIII 10 U/μL, 2× ddPCR EvaGreen Supermix (Bio-Rad, Hercules, CA, USA), and 0.1 µM primers forward and reverse. The reaction mixtures were partitioned into a droplet emulsion with the QX200 droplet generator (Bio-Rad, Hercules, CA, USA), then transferred into a 96-well PCR plate, heat sealed, and placed in a Veriti™ Thermal Cycler (Applied Biosystems, Foster City, CA, USA). The thermal cycling program was 95 °C for 10 min; 10 cycles of 94 °C for 40 s and 68 °C for 110 s, with a decrease of 0.5 °C each step; followed by 50 cycles of 94 °C for 40 s and 63 °C for 110 s; then 90 °C for 5 min, and holding at 4 °C. The PCR product was read on the QX200 droplet reader (Bio-Rad, Hercules, CA, USA), and data were analyzed with the QuantaSoft software (Bio-Rad, Hercules, CA, USA). Recombinant β-globin copy number was calculated as described in the legend of Figure 1.

### 4.6. Transient Transfection, Immunofluorescence, and RNA FISH

The cell line carrying multiple copies of a recombinant human β-globin gene was grown to ~70% confluency on MatTek (Ashland, MA, USA) #1.5 glass-bottom 35 mm plates (for live-cell microscopy) coated with poly-lysine or on Menzel-Gläser (Braunschweig, Germany) #1.5 glass coverslips (for immunofluorescence and/or RNA FISH).

Cells were transfected using Lipofectamine 2000 (Invitrogen, Waltham, MA, USA) and 600 ng of a gRNA-expressing plasmid and 400 ng of MCP-GFP or MCP-mCherry-expressing plasmid. Recombinant β-globin transcription was induced with 1 µg/mL doxycycline (Clontech, Mountain View, CA, USA) for 16–24 h. For live-cell microscopy, medium was changed to DMEM/F12 (1:1) (Gibco, New York, NY, USA), and imaging was performed.

An optimized Pol II immunofluorescence protocol was followed [47,48]. About 24 h after treatment with 1 µg/mL doxycycline, cells were fixed in 125–250 mM HEPES pH 7.6 containing 4% formaldehyde (10 min) and 8% formaldehyde (50 min) at 4ºC, followed by permeabilization in PBS pH 7.4 containing 0.5% Triton X-100 for 30 min at 4ºC with gentle rocking. Blocking was performed in PBS pH 7.4 containing 1% bovine serum albumin, 0.2% bovine skin gelatin, and 0.1% bovine milk casein for 1 h at room temperature. A rabbit anti-Pol II C-terminal domain repeats YSPTSPS phospho S2 primary antibody (IgG polyclonal; Abcam [Cambridge, UK]; catalogue number: ab5095) was incubated in a 1:1000 dilution for 2 h at room temperature in blocking solution, followed by 3x 1 h washes in the same solution. The secondary antibody (1:500–1:2000 dilution) was incubated 1 h at room temperature, and slides were rinsed overnight at 4 °C in blocking solution. Slides were washed 3× 5 min in PBS pH 7.4 0.1% Tween20, followed by 2× 5min in PBS pH 7.4 and mounted in Vectashield mounting medium (Vector Laboratories, Newark, NJ, USA).

For RNA FISH, about 24 h after treatment with 1 µg/mL doxycycline, cells were fixed in PBS pH 7.4 containing 3.7% formaldehyde for 10 min at room temperature and permeabilized in PBS PH 7.4 containing 0.5% Triton X-100 for 10 min at room temperature. They were subsequently washed 3× 5 min in PBS pH 7.4, followed by 2× 5 min in 2× saline-sodium citrate (SSC) buffer pH 7.0, 0.05% Tween20, at room temperature. Samples were blocked in 2× SSC containing 0.05% Tween20, 1% bovine serum albumin, 1 mg/mL tRNA, 2 mM vanadyl ribonucleoside complex (VRC) for 30 min at room temperature. They were hybridized overnight at 30 °C in 2× SSC containing 10% formamide, 25% dextran sulfate, and 10 nM of a FISH probe recognizing human β-globin exon 2. The FISH probe (Sigma-Aldrich, St. Louis, MO, USA) has the 5′ to 3′ sequence GCCCATAACAGCATCAGGAG and was modified with a cyanine 3 (Cy3) fluorophore at both ends of the oligonucleotide. Before hybridization, the probe was denatured for 5 min at 75 °C. After overnight incubation, coverslips were washed 2× 30 min in 2× SSC with 0.05% Tween20 and 10% formamide, followed by 2× 5 min washes in the same buffer without formamide, at 30 °C. Coverslips were further washed in PBS pH 7.4 and fixed again in PBS pH 7.4 containing 3.7% formaldehyde, washed 2× 5 min in PBS pH 7.4, and mounted in Vectashield mounting medium (Vector Laboratories, Newark, NJ, USA).

For immunofluorescence combined with RNA FISH, the FISH protocol described above was followed, with the following changes: the primary antibody was added to the hybridization solution containing probe (after denaturation and cooling down), and the secondary antibody was added during the first post-hybridization washing step, which was extended to 50 min.

### 4.7. Confocal Microscopy and FRAP

Microscopy experiments were performed at the Bioimaging Unit of Instituto de Medicina Molecular João Lobo Antunes. For simultaneous visualization of recombinant β-globin and nascent transcripts, transfected cells were visualized in a 3i Marianas spinning disk confocal microscope (Intelligent Imaging Innovations, Denver, CO, USA) equipped with a Plan-apochromat 100× 1.40 NA oil objective (Carl Zeiss, Jena, Germany). Images were recorded using an Evolve 512 EMCCD camera (Photometrics, Tucson, AZ, USA). Exposure times were 30–100 ms, and laser power was kept between 10–70%. Intensification was 200. Unless mentioned otherwise, all images shown are single-plan micrographs obtained from z-stack images with a 0.27 µm step size.

3D reconstructions were performed using the IMARIS 7.0 software (Bitplane, Zurich, Switzerland). Z-stack images of two elements among gene array, Pol II and nascent RNA, were tightly cropped around the green and red signals. Using the original micrographs as references, intensity thresholds were manually adjusted in each case in order to minimize background while keeping relative size, shape, and overlap between signals. Reconstructions were manually divided into classes based on the extent of overlap. Micrographs and 3D reconstruction images were color-corrected (red to magenta) using Adobe Photoshop CS3.

The total fluorescence intensities of the gene array, nascent RNA, and Pol II signals were determined using ImageJ 1.53e. A polygon region of interest (ROI) was placed over the fluorescent signal or adjacent nucleoplasmic background. For quantification of Pol II signal intensity, nucleoli (the only nucleoplasmic regions depleted of Pol II) were selected as the background. The integrated (total) fluorescence intensity of the signal ROI, the ROI area, and the mean fluorescence intensity of the background were determined from single-channel, maximum-intensity projection images containing only the slices where each signal was found. The total corrected fluorescence intensity of each signal was calculated as:Corrected signal intensity = integrated signal intensity − (ROI area × mean background intensity)

For FRAP experiments, cells were seeded and transfected as described above, using 1 μg of gRNA-expressing plasmid. For FRAP of dCas9-GFP bound to the recombinant β-globin operator or nucleoli, cells were co-transfected with dCas9-GFP-expressing plasmid to increase dCas9-GFP levels in the nucleoplasm and nucleoli. A Zeiss 880 point scanning confocal microscope equipped with a 1.46 NA 63× or 1.40 NA 100× oil objective (Carl Zeiss, Oberkochen, Germany) was used. A series of 40 images (frame size, 512 × 512; pixel width, 56 nm; pixel time, 1.03 μs) were acquired at regular intervals with an open pinhole. Intervals were 15 s (for dCas9-GFP bound to recombinant β-globin) or 2 s (for dCas9-GFP bound to nucleoli). Photobleaching was induced at 100% laser transmission on a circular ROI with a diameter of 30 pixels (0.8 μm radius) for 723 ms (20 iterations). For imaging, the transmission was attenuated to 3% of the bleaching intensity.

Fluorescence intensity quantification was carried out automatically for each FRAP time series using a macro created in ImageJ (https://imagej.net/ij/index.html; accessed on 30 March 2023). Briefly, time-lapse frames were background subtracted and corrected for cell displacement during image acquisition using a rigid body registration algorithm (http://bigwww.epfl.ch/thevenaz/stackreg/; accessed on 30 March 2023). The average fluorescence I(t) in the bleached region was then measured using the Time Series Analyzer plugin for ImageJ (http://rsbweb.nih.gov/ij/plugins/time-series.html; accessed on 30 March 2023), taking advantage of the Recenter function to automatically coincide the geometric center of the measuring ROI with the center of mass of the green-labeled transcription sites, thus correcting for any minor intranuclear movement of the gene array. In each time frame, the nuclear region was subsequently identified using k-means clustering (https://github.com/ij-plugins/ijp-toolkit/wiki/k%E2%80%90means-Clustering; accessed on 30 March 2023), and the average fluorescence in the nucleus T(t) at time t after bleaching was measured to normalize FRAP recovery curves, as previously described [49].

### 4.8. ChIP and qPCR Analysis

A previously published protocol [50] was followed. Cells stably expressing a single copy of a recombinant human β-globin gene, dCas9-GFP, Tet-ON 3G, and a gRNA-recognizing telomeric repeat, β-globin exon 1, exon 2, or exon 3, were grown on 10-cm plates. When confluency reached 50–70%, recombinant human β-globin expression was induced with 1 µg/mL doxycycline. About 24 h later, chromatin elements were cross-linked with 1% formaldehyde (diluted in PBS pH 7.4) for 10 min at room temperature, followed by quenching with 140 mM glycine for 5 min at room temperature. Cells were scraped, harvested in PBS pH 7.4, and resuspended in 700 µL 1% SDS, 50 mM Tris-HCl, 10 mM EDTA, pH 8.0, supplemented with 1/8 tablet/mL Complete, EDTA-free Protease Inhibitor Cocktail (Roche, Basel, Switzerland). After an incubation period of 10–30 min on ice, cell suspensions were sonicated 5× 20 s in Soniprep 150 (Sanyo, Osaka, Japan) operating at 10 micron amplitude. The lysate was cleared by centrifugation, and a 50 µL aliquot was taken for total DNA purification (input).

A total of 8 µL samples of chromatin fragments (either before or after sonication) were analyzed in a 2% agarose gel stained with Midori Green (Nippon Genetics, Tokyo, Japan). 1 kbp Plus DNA Ladder (Thermo Fisher Scientific, Waltham, MA, USA) was used as a molecular mass marker.

For immunoprecipitation, the leftover was 2-fold diluted with dilution buffer (167 mM NaCl, 17 mM Tris-HCl, 1.1% Triton X-100, 1.2 mM EDTA, 0.01% SDS, pH 8.0) and incubated with 20 µL suspension of magnetic Dynabeads^TM^ Protein G (Invitrogen, Waltham, MA, USA) for 1 h at 4 °C with rotation. After Dynabeads harvesting in a magnetic rack, each sample was split in two and further diluted to 1.3 mL with dilution buffer. Dilutions were incubated overnight at 4 °C with rotation in the presence of 3 µg mouse anti-GFP (from IgG1k clones 7.1 and 13.1; Roche [Basel, Switzerland]; catalogue number 11814460001) or 3 µg rabbit anti-Pol II C-terminal domain repeats YSPTSPS phospho S5 (IgG polyclonal; Abcam [Cambridge, UK]; catalog number: ab5131) antibodies. After incubation, 50 µL dynabead suspension were added to each sample and incubated for 3 h at 4 °C with rotation and subjected to four washes. First wash was in 20 mM Tris-HCl pH 8.0, 0.1% SDS, 1% Triton X-100, 2 mM EDTA containing 150 mM NaCl. The second wash was in the same buffer containing 500 mM NaCl. The third wash was in 0.25 mM LiCl, Tris-HCl, pH 8.0, 1% Nonidet P-40, 1% sodium deoxycholate, 1 mM EDTA. The fourth wash was in 10 mM Tris-HCl pH 8.0, 1 mM EDTA. Buffer was completely removed, and Dynabeads boiled for 10 min in a 10% Chelex-100 (Bio-Rad, Hercules, CA, USA) suspension, incubated for 45 min at 55 °C with 20 µg proteinase K (Thermo Fisher Scientific, Waltham, MA, USA), and re-boiled for 10 min. Immunoprecipitated DNA fragments were collected from the supernatant in a total volume of 160 µL.

For total DNA purification, 50 µL sample was added 450 µL dilution buffer and 20 µL 5 M NaCl and incubated overnight at 65 °C. 10 µL EDTA, 20 µL Tris-HCl pH 6.8, 20 µg proteinase K (Thermo Fisher Scientific, Waltham, MA, USA) were added and incubated 90 min at 45 °C. Total DNA was purified by extraction in phenol-chloroform-isoamyl alcohol (Thermo Fisher Scientific, Waltham, MA, USA) and chloroform partitioning, followed by precipitation and dissolution in 50 µL DNase/RNase-free water.

Samples and 5-fold diluted inputs were analyzed by quantitative PCR using 5 µL sample or diluted input and 10 µL of a mixture containing iTaq Universal SYBR Green Supermix (Bio-Rad, Hercules, CA, USA) and the primer pairs listed in Appendix A. Amplification was monitored in real time during 40 cycles using SYBR Green standard protocol. Each ChIP sample (or 5-fold diluted input) was amplified in triplicate, and the corresponding cycle threshold (Ct) was averaged. Then, the averaged Ct of each diluted input was subtracted from the averaged Ct of the corresponding dCas9 and Pol II samples. The ΔCt value of each sample was obtained. The fold-over input of each sample was calculated as 2^^(−ΔCt)^. Additional ChIP assays were performed, and results were combined.

### 4.9. Isolation of Nascent RNA and ddRT-PCR

Cells stably expressing dCas9-GFP and a gRNA-recognizing telomeric repeat, β-globin exon 1, exon 2, or exon 3, were cultured in 10-cm plates. Recombinant β-globin expression was induced overnight with 1 µg/mL doxycycline. Cells were harvested, resuspended in culture medium, distributed into two aliquots, centrifuged, and resuspended in PBS pH 7.4. The fractionation procedure was performed at 4 °C. Samples were centrifuged 3 min at 1500× *g*, and pellets resuspended in 1 mL 10 mM Tris pH 7.4, 10 mM NaCl, 3 mM MgCl_2_. After centrifuging 3 min at 800× *g*, pellets were resuspended in 200 µL 10 mM Tris-HCl pH 7.4, 10 mM NaCl, 3 mM MgCl_2_, 10% glycerol, 0.5% Nonidet P-40, 0.5 mM dithiothreitol (DTT), 100 units/mL RNAse inhibitor (Bioline, Toronto, ON, Canada) and centrifuged 3 min at 4500× *g*. Pellets were resuspended in 25 µL 20 mM tris-HCl pH 8.0, 75 mM NaCl, 0.5 mM EDTA, 0.85 mM DTT, 0.2125 mM PMSF, 50% glycerol, 100 units/mL RNase inhibitor (Bioline, Toronto, ON, Canada). 225 µL 10 mM HEPES pH 7.6, 300 mM NaCl, 0.2 mM EDTA, 1 mM DTT, 7.5 mM MgCl_2_, 1 M urea, 1% Nonidet P-40, 100 units/mL RNase inhibitor (Bioline, Toronto, ON, Canada) were added. Samples were centrifuged at 16,000× *g* for 2 min and resuspended in 200 µL 10 mM Tris-HCl pH 7.5, 500 mM NaCl, 10 mM MgCl_2_, 100 units/mL DNaseI (Roche, Basel, Switzerland), 100 units/mL RNase inhibitor (Bioline, Toronto, ON, Canada), and incubated 30 min at 37 °C. Chromatin-associated RNA was extracted with PureZOL RNA isolation reagent (Bio-Rad, Hercules, CA, USA), followed by chloroform partitioning, and dissolved in 15 µL RNase-free water.

For ddRT-PCR, One-Step RT-ddPCR Advanced Kit for Probes (Bio-Rad, Hercules, CA, USA) was used. For each condition, technical duplicates were prepared by mixing on ice 10 µL Supermix, 4 µL reverse transcriptase, 2 µL 300 mM DTT, and 4 µL of each primer and probe set. Primers and probes used in ddRT-PCR are listed in Appendix A. 20 µL chromatin-associated RNA diluted to 0.25 ng/µL were added to each mix (final volume: 40 µL), which was split into two aliquots. Droplets were prepared using 20 µL mix and 70 µL Droplet Generation Oil for Probes (Bio-Rad, Hercules, CA, USA). Thermal cycling conditions included reverse transcription (60 min at 50 °C), enzyme activation (10 min at 95 °C), followed by 50 cycles of denaturation (30 s at 95 °C), and annealing/extension (1 min and 40 s at 65 °C). Finally, enzymes were heat-inactivated (10 min at 98 °C). After droplet reading, results were visualized in QuantaSoft 1.7.4.0917 software, where the threshold between negative (void of amplified target) and positive (containing at least one amplified target copy) droplets was manually adjusted. For each reaction, the number of copies/µL for both β-globin and GAPDH was calculated by the QuantaSoft software (Bio-Rad, Hercules, CA, USA). Results from each technical duplicate were combined, and levels of recombinant β-globin nascent RNA were normalized by dividing by the corresponding number of GAPDH RNA copies. Additional ddRT-PCR assays were performed, and results were combined.

### 4.10. Statistics

Data processing calculations, coefficients of determination, as well as line and pie graphs, were obtained in Microsoft Excel 2016. Scatter plots and statistical analyses were performed in GraphPad Prism 8.0.2 (GraphPad Software, San Diego, CA, USA). In each cell line, the recombinant human β-globin gene copy number was determined as the mean ± standard deviation of the copy numbers obtained using each of the two stem-loop-specific primer pairs (Figure 1C,D). In all confocal microscopy assays, each experimental measurement corresponds to a single cell or a single 3D reconstruction. In ChIP and ddRT-PCR assays, each measurement corresponds to the cell population from a single cell growth plate. FRAP quantification data were represented as the mean ± standard deviation. All other numerical data were assumed to have a non-Gaussian distribution and represented as the median ± interquartile range. In 3D reconstructions, ChIP, and ddRT-PCR data, statistical significance was assessed using the Mann–Whitney test, in which one-tailed (ChIP) or two-tailed (3D reconstructions and ddRT-PCR) *p*-values < 0.05 were considered statistically significant.

## Figures and Tables

**Figure 1 ijms-25-08411-f001:**
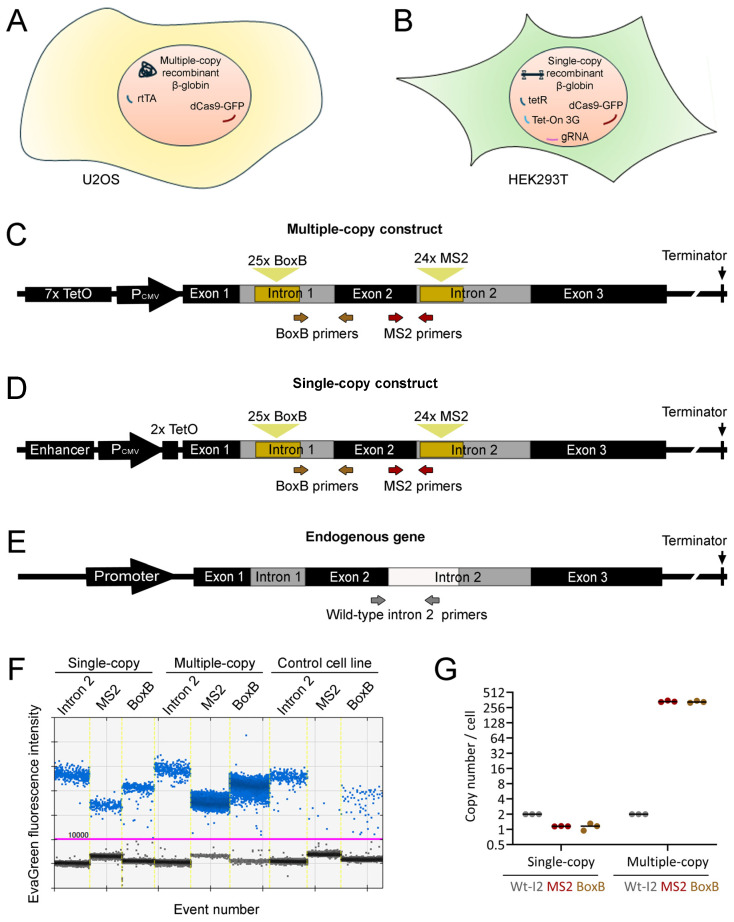
Cell lines used in the present study. (**A**,**B**) Schematic representations of the cell lines carrying (**A**) multiple copies or (**B**) a single copy of a recombinant human β-globin gene. Transgene expression is inducible by doxycycline. (**A**) The U2OS Tet-On-based multiple-copy integration of the β-globin construct is a monoclonal cell line that also contains transgenes coding for the tetracycline transactivator protein (rtTA) and catalytically dead Cas9 fused to the green fluorescent protein (dCas9-GFP). (**B**) The HEK293T-based cell line with a single copy integration of the β-globin construct in the recombination site also contains a tetracycline repressor (tetR) gene and additional genes coding for dCas9-GFP, tet-On 3G, and a guide RNA (gRNA) targeting whether telomeric DNA or a specific region in the human β-globin gene. (**C**,**D**) Schematic representations of the recombinant β-globin gene construct carrying arrays of 25 and 24 stem-loop coding repeats derived from the ʎN bacteriophage (BoxB; intron 1) and the MS2 bacteriophage (MS2; intron 2), respectively, and a deletion in intron 2. Their promoter regions are different. (**C**) Promoter for expression in cell line (**A**) contains seven copies of the tetracycline-induced operator (TetO) upstream of the CMV-derived minimal promoter sequence. (**D**) The regulatory sequence for expression in cell line (**B**) contains an enhancer upstream, the CMV promoter, and two copies of the TetO operator downstream, which bind the Tet-repressor in the absence of doxycycline. (**E**) Representation of the endogenous β-globin gene, highlighting the region in intron 2 that is deleted in the recombinant constructions (in white). (**F**,**G**) Quantification of recombinant β-globin gene copy number by droplet digital PCR (ddPCR). (**F**) Representative ddPCR raw data. The primer pairs used to amplify unique sequences from BoxB and MS2 insertion regions are depicted as arrows in panels (**C**,**D**), and the primer pair for the endogenous gene, present as two copies per genome, is depicted in panel (**E**). Each dot in the graph represents the fluorescence intensity result of PCR amplification in a single droplet. Droplets with fluorescence higher than 1000 (magenta line) were assigned positive (blue), while nonfluorescent droplets were assigned negative (black). In the multiple-copy cell line, MS2 and BoxB regions are revealed to be very abundant. In the non-transfected U2OS cell line (used as a negative control), MS2 and BoxB detection are barely negative and were assigned as background. (**G**) Copy number per cell was calculated as the ratio to endogenous β-globin number of copies times 2, after background subtraction. In the single-copy cell line (**B**), the copy number of the MS2 region is 1.16 ± 0.09, and that of the BoxB region is 1.15 ± 0.25. As a whole, the copy number of the recombinant gene was found to be 1.15 ± 0.17. In the multiple-copy monoclonal cell line (**A**), the copy number of the MS2 regions is 338.7 ± 13.5, and that of the BoxB regions is 332.6 ± 16.9. Together, the copy number of the inserted recombinant gene was determined to be 335.6 ± 15.3. (Wt-I2, wild-type intron 2).

**Figure 2 ijms-25-08411-f002:**
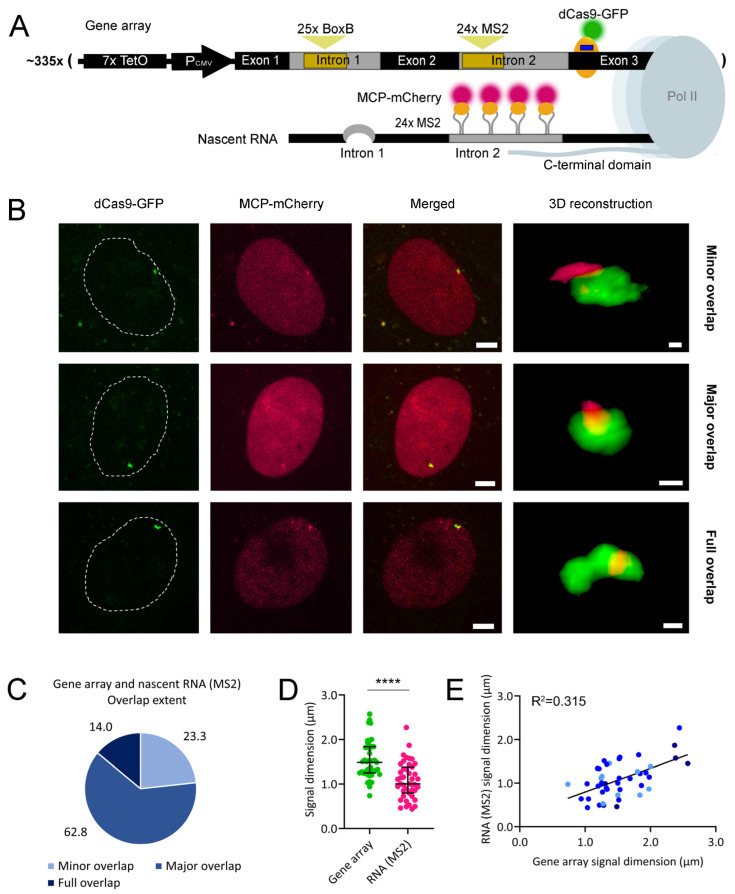
Visualization of an array of ~335 recombinant human β-globin genes and their MS2-labeled nascent RNA in living U2OS cells. (**A**) Schematic representation of one gene copy (detailed in Figure 1C) from the array, which is modified with sequences coding for arrays of 25 BoxB stem-loops and 24 MS2 stem-loops in introns 1 and 2, respectively. dCas9 (yellow) fused to GFP (green) is bound to exon 3. The dCas9 guide RNA (gRNA; blue bar) is hybridized with the template strand of the gene. In its nascent RNA, MS2 stem-loops are labeled with a fusion of MS2 coat protein (MCP; yellow) and mCherry (magenta). Looping of intron 1 represents its co-transcriptional splicing. (**B**) Representative examples of spinning-disk confocal micrographs and corresponding 3D reconstructions showing the fluorescently labeled human β-globin gene array (green) and its MS2-labeled nascent RNA (magenta). The dashed line represents the nuclear membrane. Scale bars correspond to 5 µm and 0.3 µm in single-plan micrographs and 3D reconstructions, respectively. (**C**) Quantification of the overlap extent in the dataset, where the percentages of each extent type are indicated near the pie graph. (**D**) Quantification of the length of the gene array and MS2-labeled nascent RNA signals in the dataset. Statistical significance was assessed with the Mann–Whitney test (****: *p* < 0.0001). (**E**) Spearman correlation analysis between the signal lengths of the gene array and MS2-labeled nascent RNA of data points from panel (**D**). The coefficient of determination is indicated. A linear regression curve was fitted to the data points. Data points in panel (**E**) are colored according to their overlap extent in panel (**C**). Results from panels (**C**–**E**) were compiled from 43 3D reconstructions of 43 single cells from ten independent experiments. Three of these reconstructions are shown in panel (**B**); the other 40 3D reconstructions are shown in Appendix A.

**Figure 3 ijms-25-08411-f003:**
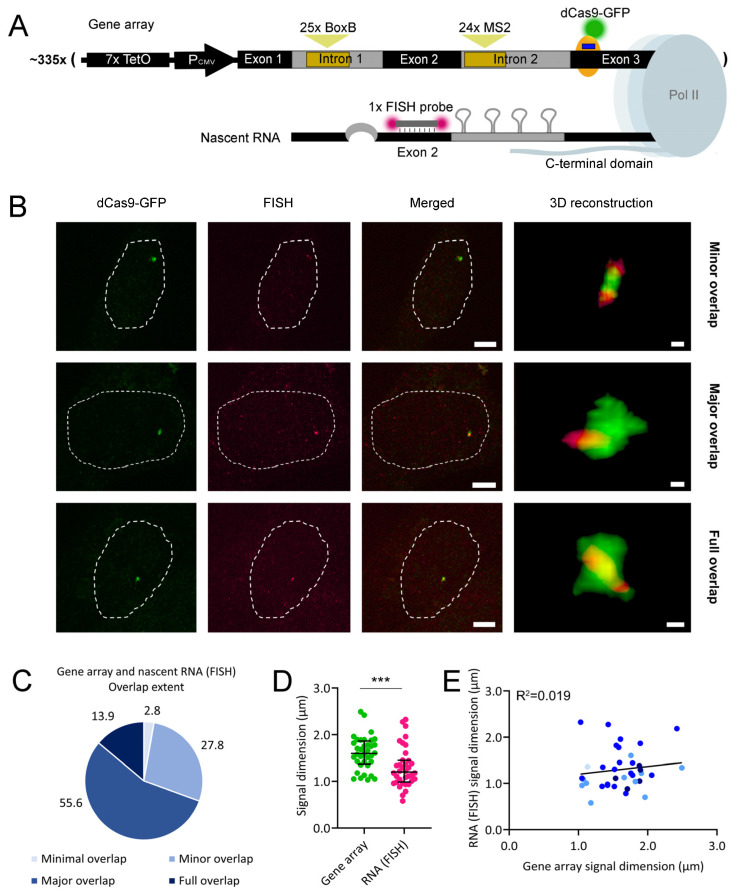
Visualization of an array of recombinant human β-globin genes and their FISH-labeled nascent RNA in fixed U2OS cells. (**A**) Schematic representation of one gene copy from the array, which is modified with sequences coding for arrays of 25 BoxB stem-loops and 24 MS2 stem-loops in introns 1 and 2, respectively. dCas9 (yellow) fused to GFP (green) is bound to exon 3. The gRNA (blue bar) is hybridized with the template strand of the gene. Its nascent RNA is labeled with a fluorescence in situ hybridization (FISH) probe containing one fluorophore (magenta) at each end of the oligonucleotide. The FISH probe has only one binding site per nascent RNA molecule, where looping of intron 1 represents its co-transcriptional splicing. (**B**) Representative examples of spinning-disk confocal micrographs and corresponding 3D reconstructions, showing the gene array (green) and corresponding FISH-labeled nascent RNA (magenta). The dashed line represents the nuclear membrane. Scale bars correspond to 5 µm and 0.3 µm in single-plan micrographs and 3D reconstructions, respectively. (**C**) Quantification of the overlap extent in the dataset, where the percentages of each extent type are indicated near the pie graph. (**D**) Quantification of the length of the gene array and FISH-labeled nascent RNA signals in the dataset. Statistical significance was assessed with the Mann–Whitney test (***: *p* = 0.0006). (**E**) Spearman correlation analysis between the signal lengths of the gene array and FISH-labeled nascent RNA of data points from panel (**D**). The coefficient of determination is indicated. A linear regression curve was fitted to the data points. Data points in panel (**E**) are colored according to their overlap extent in panel (**C**). Results from panels (**C**–**E**) were compiled from 36 3D reconstructions of 35 cells from two independent experiments. Three of these reconstructions are shown in panel B; the other 33 3D reconstructions are shown in Appendix A.

**Figure 4 ijms-25-08411-f004:**
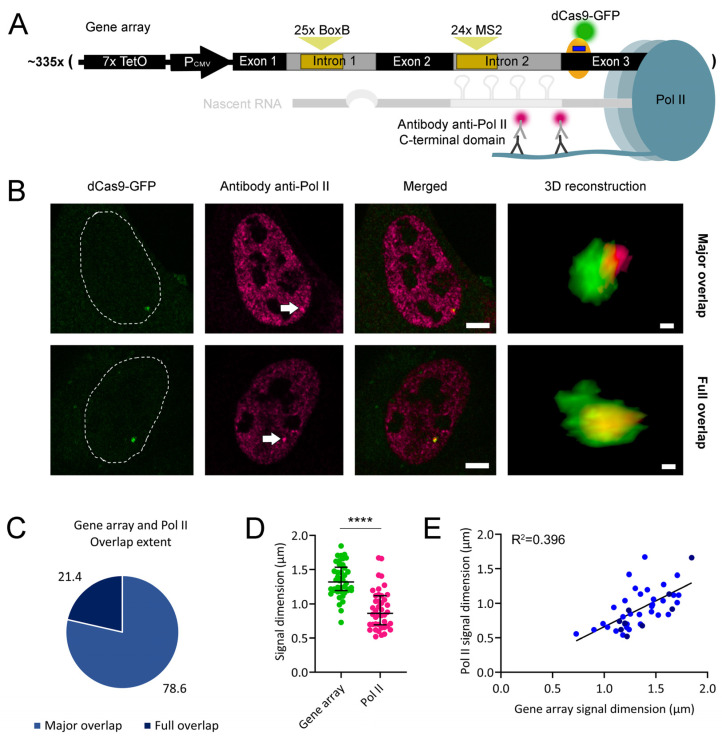
Visualization of an array of recombinant human β-globin genes and transcribing Pol II in fixed U2OS cells. (**A**) Schematic representation of one gene copy from the array, which is modified with sequences coding for arrays of 25 BoxB stem-loops and 24 MS2 stem-loops in introns 1 and 2, respectively. A fusion of dCas9 (yellow) and GFP (green) is bound to exon 3. The gRNA (blue bar) is hybridized with the template strand of the gene. A fluorescently labeled secondary antibody is bound to a primary antibody recognizing the C-terminal domain repeats of elongating RNA polymerase II (Pol II). (**B**) Representative examples of spinning-disk confocal micrographs and corresponding 3D reconstructions showing the fluorescently labeled gene copies (green) and associated Pol II (magenta). The dashed line represents the nuclear membrane. Pol II signals are indicated by white arrows. Scale bars correspond to 5 µm and 0.3 µm in single-plan micrographs and 3D reconstructions, respectively. (**C**) Quantification of the overlap extent in the dataset, where the percentages of each extent type are indicated near the pie graph. (**D**) Quantification of the length of the gene array and associated Pol II signals in the dataset. Statistical significance was assessed with the Mann–Whitney test (****: *p* < 0.0001). (**E**) Spearman correlation analysis between the signal lengths of gene array and Pol II of data points from panel (**D**). The coefficient of determination is indicated. A linear regression curve was fitted to the data points. Data points in panel (**E**) are colored according to their overlap extent in panel (**C**). Results from panels (**C**–**E**) were compiled from 42 3D reconstructions of 41 single cells from four independent experiments. Two of these reconstructions are shown in panel B; the other 40 3D reconstructions are shown in Appendix A.

**Figure 5 ijms-25-08411-f005:**
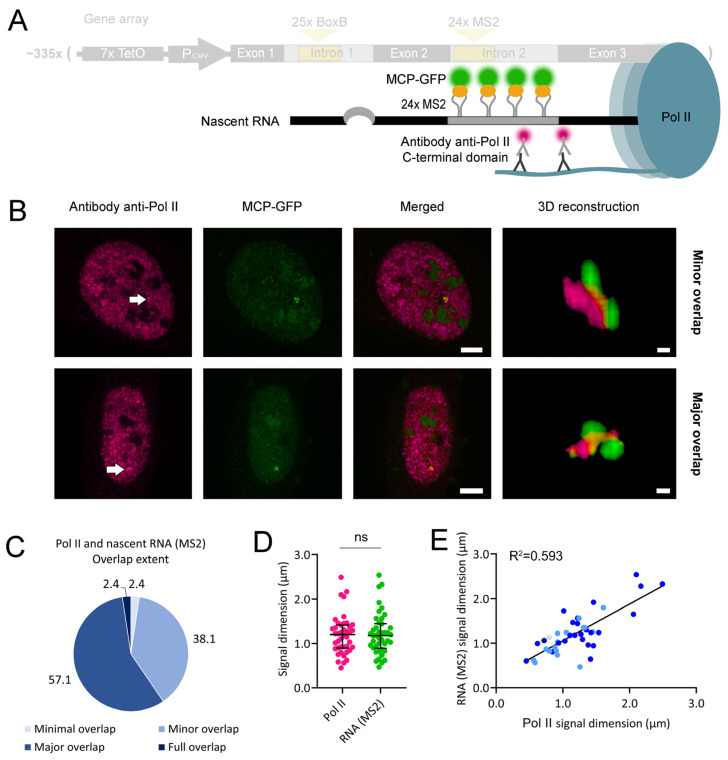
Visualization of Pol II and associated MS2-labeled nascent RNA transcribed from an array of recombinant human β-globin genes in fixed U2OS cells. (**A**) Schematic representation of a nascent RNA molecule transcribed from the gene array, in which the 24 MS2 stem loops inserted into intron 2 of the nascent RNA are labeled with an MCP-GFP fusion protein. Pol II is bound to antibodies recognizing its C-terminal domain, which are further bound to fluorescently labeled secondary antibodies. (**B**) Representative examples of spinning-disk confocal micrographs and corresponding 3D reconstructions showing fluorescently labeled Pol II (magenta) and MS2-labeled nascent RNA (green). Dashed lines represent the nuclear membrane. Pol II signals are indicated by white arrows. Scale bars correspond to 5 µm and 0.3 µm in single-plan micrographs and 3D reconstructions, respectively. (**C**) Quantification of the overlap extent in the dataset, where the percentages of each extent type are indicated near the pie graph. (**D**) Quantification of the length of the Pol II and MS2-labeled nascent RNA signals in the dataset. Statistical significance was assessed with the Mann–Whitney test (ns: non-significant). (**E**) Spearman correlation analysis between the signal lengths of Pol II and MS2-labeled nascent RNA of data points from panel (**D**). The coefficient of determination is indicated. A linear regression curve was fitted to the data points. Data points in panel (**E**) are colored according to their overlap extent in panel (**C**). Results from panels (**C**–**E**) were compiled from 42 3D reconstructions of 42 single cells from three independent experiments. Two of these reconstructions are shown in panel B; the other 40 3D reconstructions are shown in Appendix A.

**Figure 6 ijms-25-08411-f006:**
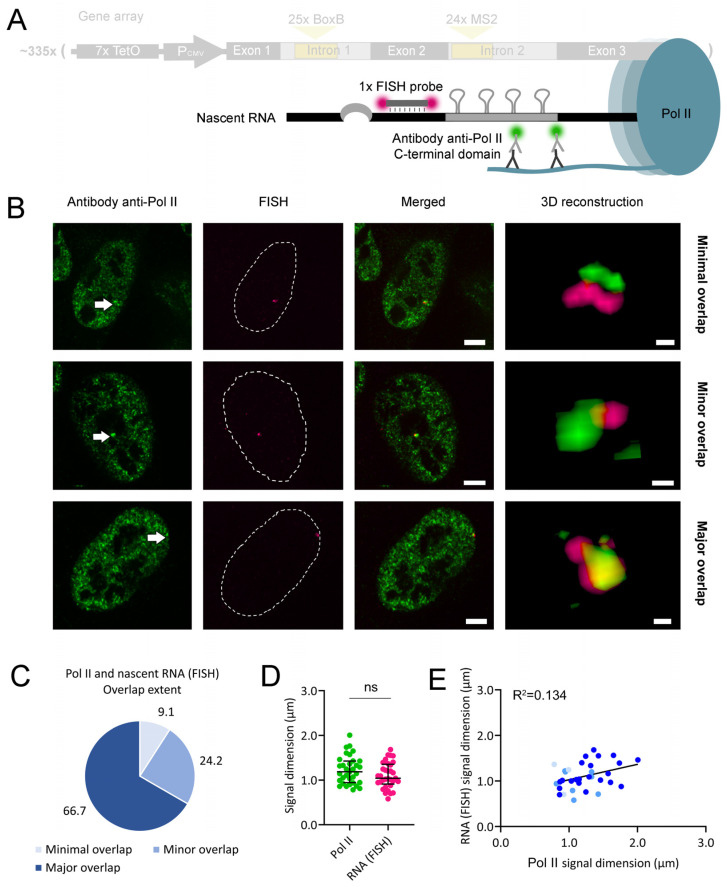
Visualization of Pol II and associated FISH-labeled nascent RNA transcribed from an array of recombinant human β-globin genes in fixed U2OS cells. (**A**) Schematic representation of a nascent RNA molecule transcribed from the gene array, in which exon 2 is labeled with a FISH probe containing one fluorophore at each end of the oligonucleotide. The FISH probe has only one binding site per nascent RNA molecule. Pol II is bound to antibodies recognizing its C-terminal domain, which are further bound to fluorescently labeled secondary antibodies. (**B**) Representative examples of spinning-disk confocal micrographs and corresponding 3D reconstructions showing fluorescently labeled Pol II (green) and FISH-labeled nascent RNA (magenta). Dashed lines represent the nuclear membrane. Pol II signals are indicated by white arrows. Scale bars correspond to 5 µm and 0.3 µm in single-plan micrographs and 3D reconstructions, respectively. (**C**) Quantification of the overlap extent in the dataset. (**D**) Quantification of the length of the Pol II and FISH-labeled nascent RNA signals in the dataset, where the percentages of each extent type are indicated near the pie graph. Statistical significance was assessed with the Mann–Whitney test (ns: non-significant). (**E**) Spearman correlation analysis between the signal lengths of Pol II and FISH-labeled nascent RNA of data points from panel (**D**). The coefficient of determination is indicated. A linear regression curve was fitted to the data points. Data points in panel (**E**) are colored according to their overlap extent in panel (**C**). Results from panels (**C**) to (**E**) were compiled from 33 3D reconstructions of 33 single cells from two independent experiments. Three of these reconstructions are shown in panel B; the other 30 3D reconstructions are shown in Appendix A.

**Figure 7 ijms-25-08411-f007:**
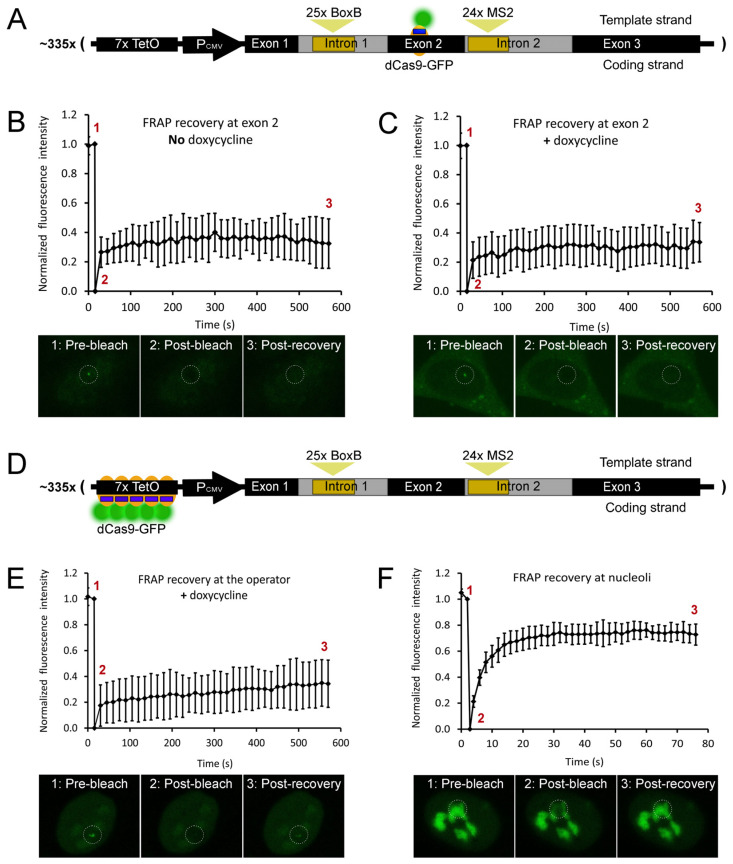
Fluorescence recovery after photobleaching (FRAP) of dCas9-GFP bound to an array of recombinant human β-globin genes in U2OS cells. (**A**) Schematic representation of one gene copy from the array (related to Appendix A), which is modified with sequences coding for arrays of 25 BoxB stem-loops and 24 MS2 stem-loops in introns 1 and 2, respectively. dCas9 (yellow) fused to GFP (green) is bound to exon 2, with the gRNA (blue bar) hybridized with the template strand of the gene. (**B**,**C**) Quantified fluorescence intensity of dCas9-GFP bound to exon 2 of the recombinant gene before and after photobleaching in (**B**) the absence (13 cells from eight independent experiments) or (**C**) the presence of 1 µg/mL doxycycline (16 cells from seven independent experiments). (**D**) Schematic representation of the construct shown in (**A**), in which dCas9 is bound to each of the seven tetracycline-induced operator (TetO) copies and the gRNA is hybridized with the coding strand of the gene. (**E**,**F**) Quantified fluorescence intensity of dCas9-GFP bound to (**E**) the TetO operator of the recombinant gene in the presence of 1 µg/mL doxycycline (25 cells from seven independent experiments) or (**F**) nonspecifically bound to nucleoli (16 cells from four independent experiments). Results are expressed as mean ± standard deviation. In the graphs, the numbers indicate the pre-bleach (1), post-bleach (2), and post-recovery (3) stages. Snapshots of each stage from representative FRAP videos are shown under the graphs in panels B (Appendix A), C (Appendix A), E (Appendix A), and F (Appendix A). In each snapshot, the photobleached region of interest is indicated with a dotted circle.

**Figure 8 ijms-25-08411-f008:**
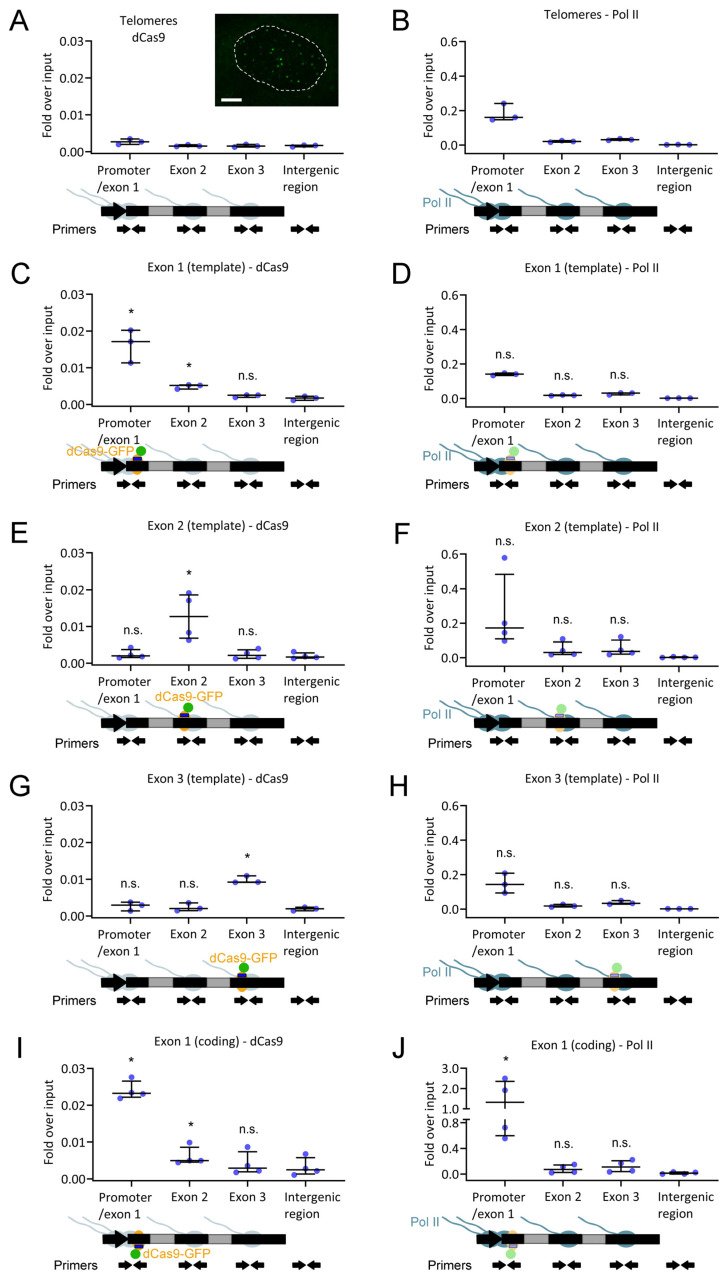
Levels of transcriptionally paused Pol II in the promoter region of a dCas9-bound recombinant human β-globin gene in HEK293T cells. (**A**–**J**) Chromatin immunoprecipitation folds over input graphs of dCas9 enrichment in the promoter/exon 1, exon 2, and exon 3 of the recombinant β-globin gene and an intergenic region of the genome. Here, dCas9 was targeted to (**A**,**B**) telomeric DNA repeats, (**C**,**D**) exon 1, (**E**,**F**) exon 2, and (**G**,**H**) exon 3 of the recombinant gene (gRNA hybridized with template strand). In panels (**I**,**J**), dCas9 was bound to exon 1, with the gRNA hybridizing the coding strand of the gene. The panel A inset is a representative maximum intensity projection image of dCas9-GFP-labeled telomeric repeats in U2OS cells (the dashed line represents the nuclear membrane; the scale bar corresponds to 5 µm). In all panels, each graph contains a schematic representation of the recombinant gene structure (exons: black; introns: gray) with the respective dCas9-GFP or Pol II distributions. Quantitative PCR amplification regions are indicated with arrows in the corresponding gene regions. Results represent the median and interquartile ranges of (**A**–**D**,**G**,**H**) three or (**E**,**F**,**I**,**J**) four independent cell growth plates. Statistical significance was assessed with the Mann–Whitney test (*: *p* < 0.05). In panels C, E, G, and H, the levels of bound dCas9 were compared with those in the same gene region in panel (**A**). In panels D, F, H, and J, the levels of transcriptionally paused Pol II in each gene region are compared with those in the same region in panel (**B**).

**Figure 9 ijms-25-08411-f009:**
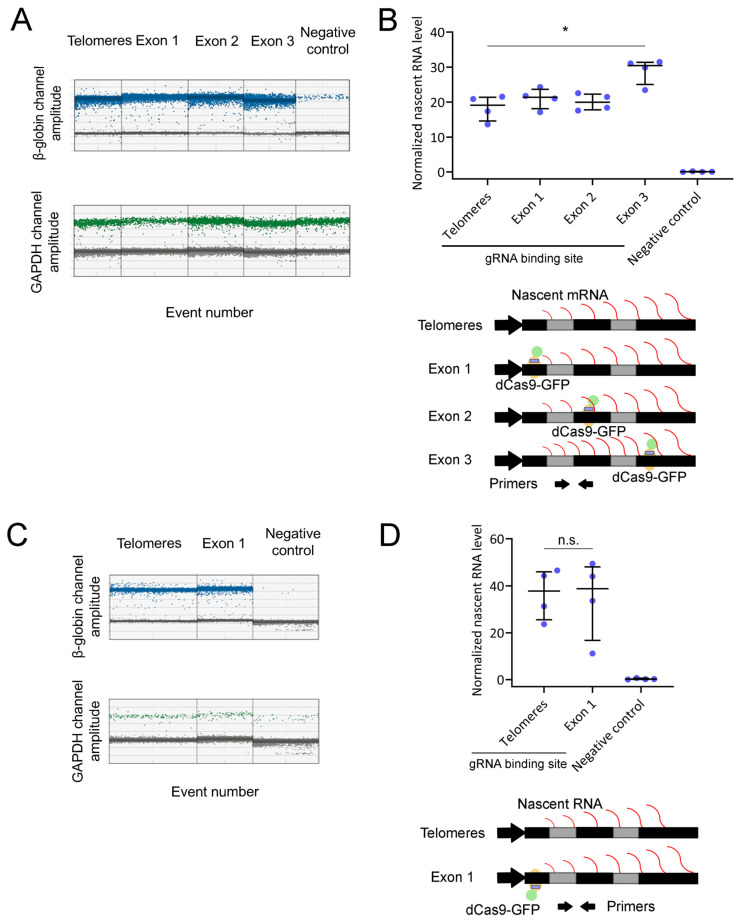
Quantification of nascent RNA transcribed from a dCas9-bound recombinant human β-globin gene in HEK293T cells. (**A**) Representative example of droplet digital reverse transcription PCR (ddRT-PCR) raw data depicting droplets containing none or at least one amplified copy of cDNA from nascent RNA, either from β-globin (**top**) or GAPDH (**down**). Positive droplets were colored blue for β-globin and green for GAPDH, while negative droplets are represented in gray. Here, dCas9 was targeted to telomeric DNA or exons 1–3 of the recombinant β-globin gene (gRNA hybridized with template strand). As a negative control, the parental cell line (HEK293T cells without the recombinant gene) was also included. (**B**) GAPDH-normalized levels of nascent RNA levels of recombinant β-globin determined from (**A**). Under the graph, interpretive models represent potential nascent RNA distribution profiles throughout the recombinant gene body for dCas9-GFP bound to different gene regions (exons: black; introns: gray). (**D**) Representative example of ddRT-PCR raw data (similar to panel A). Here, dCas9 was targeted to telomeric DNA, or exon 1, of the recombinant β-globin gene (gRNA hybridized with coding strand). (**D**) GAPDH-normalized levels of nascent β-globin mRNA determined from (**C**). Under the graph, interpretive models represent potential nascent RNA distribution profiles throughout the recombinant gene body for gRNA-dCas9-GFP bound to β-globin exon 1 (exons: black; introns: gray). Results represent the median and interquartile ranges of four independent cell growth plates. Statistical significance was assessed with the Mann–Whitney test (*: *p* < 0.05).

**Figure 10 ijms-25-08411-f010:**
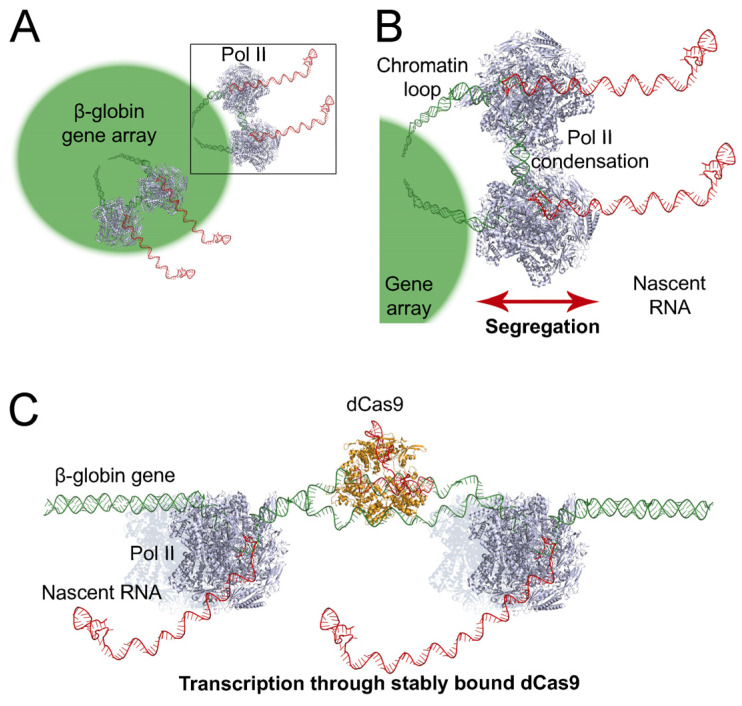
Interpretive model. (**A**) In an array of ~335 recombinant β-globin genes, chromatin loops are pulled out by transcribing Pol II. These loops may be formed from central or peripheral regions of the array. The latter case is indicated with a square and includes the loop, Pol II, and its nascent RNA. (**B**) Magnification of the chromatin loop highlighted in panel (**A**). Pol II biomolecular condensation can potentially drive loop formation, inducing segregation among the gene array, Pol II, and nascent RNA. (**C**) Pol II can maintain its transcriptional activity in the presence of dCas9 stably bound to a gene. For simplicity, co-transcriptional splicing, the Pol II C-terminal domain, and additional molecules involved in mRNA transcription and processing are not represented. Protein and nucleic acid structure images were generated with PyMOL (Schrödinger, Inc., New York, NY, USA) using structural elements deposited at the Protein Data Bank under the accession codes 4OO8, 5FLM, and 7DMQ.

## Data Availability

The data that support the findings of this study are included as Appendix A. These data include the full collection of 3D reconstructions (Appendix A) and a Microsoft Excel file containing the numerical data used in Figure 1, Figure 2, Figure 3, Figure 4, Figure 5, Figure 6, Figure 7, Figure 8 and Figure 9 and Appendix A. Original microscopy images and videos are available from the corresponding author upon reasonable request.

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
