# Peer review of "Human RNA Polymerase II Segregates from Genes and Nascent RNA and Transcribes in the Presence of DNA-Bound dCas9"

_ijms, 2024, doi:10.3390/ijms25158411_

Round 1

Reviewer 1 Report

Comments and Suggestions for Authors

This manuscript entitled “Human RNA polymerase II segregates from genes and nascent RNA and transcribes in the presence of DNA-bound dCas9” from João Pessoa and Célia Carvalho uses confocal microscopy to visualize a gene array, its transcript and RNA polymerase II during transcription. They also study how RNA polymerase II proceeds with transcription through a physical block such as dCas9 bound to DNA.  Using this approach, they conclude that a single dCas9 is not sufficient to efficiently block transcription.

The aim of the paper is interesting and the results are clearly presented. However, the last section of the paper would require more experiments (see below for more details) to fully support their conclusions.

Detailed comments : 

Section 2.1. Recombinant human cell lines to study in vivo transcription

It would be interesting to know the insertion site of the transgene(s) in the U2OS and HEK293T cells.   

Section 2.2. Human β-globin nascent RNA segregates from its originating genes 

Did the author measure the signal for the gene array in the absence of transcription, before adding the doxycycline? This control would be interesting to know how transcription could modify the structure of the gene array. 

I understand that the live-cell microscopy was performed 16-24 hours after the transcription was induced. This seems a really long time knowing that previous studies (see for example Janicky et al, Cell. 2004,116(5): 683–698), showed that transcription started as early as 10 minutes after addition of doxycycline. Did the authors performed some kinetics to test how the signal ratio was changing over time? This experiment could be a nice addition to the paper. 

Also, can the authors indicate how long after adding the doxycycline were the cells fixed for the RNA FISH experiment?

When the authors compare the signal for the gene array and the transcript (fig 2 and 3), they consider the size of the signal and measure the longest diameter of the signal. It would be interesting to repeat the quantification using the intensity of the signal as it should reflect better the level of RNA. This comment is valid for all sections of the paper.

Section 2.3. mRNA transcription is not homogeneous among identical human β-globin genes

Can the authors mention how long after adding the doxycycline were the cells fixed for the Pol II immunofluorescence experiment?

By using an antibody directed at the phosphorylated CTD, the authors are only considering the elongating polymerases at the time the cells were fixed.  It would be interesting to test the total amount of polymerases recruited to the gene array.

Section 2.4. Human β-globin nascent RNA segregates from RNA Polymerase II 

It would be interesting to test if the overlap between RNA polymerase II and its transcript is more important shortly after transcription was induced.

Section 2.5. dCas9 binding to a human β-globin gene is stable over time

The authors conclude this section by saying :  “The stable binding of dCas9-GFP to exon 2 of the β-globin gene under high levels of  Pol II transcription (video 2; Figure 7B) suggests that dCas9 might be able to block mRNA transcriptional elongation”  but they do not  know at this point if the RNA polymerase is blocked or if it can pass through the obstacle. I think this sentence should be modified.

Section 2.6. RNA polymerase II transcribes in the presence of DNA-bound dCas9 

In this section, the authors perform Pol II ChIP using an antibody directed against Pol II CTD phosphorylated at ser5. This modification is mostly found in polymerase at the beginning of the elongation (when the first few hundreds NT are transcribed, and is progressively replaced by ser2 phosphorylation. Knowing this, it is not surprising to find RNA Pol II only in promoter and exon I and not in exon 2 and 3. To be really conclusive, this experiment should be repeated using an antibody recognising Pol II CTD phosphorylated at ser2 or ser2 and ser5.

Also, could the authors indicate the average size of the chromatin fragments after sonication? 

In the discussion, the authors mention that they can observe some transcription in the absence of doxycycline. Could they give more details on the level of transcription observed before induction?  Could this affect the interpretation of their last experiment and explain why they observe some transcripts in the presence of dcas9 ?

Author Response

This manuscript entitled “Human RNA polymerase II segregates from genes and nascent RNA and transcribes in the presence of DNA-bound dCas9” from João Pessoa and Célia Carvalho uses confocal microscopy to visualize a gene array, its transcript and RNA polymerase II during transcription. They also study how RNA polymerase II proceeds with transcription through a physical block such as dCas9 bound to DNA. Using this approach, they conclude that a single dCas9 is not sufficient to efficiently block transcription.

The aim of the paper is interesting and the results are clearly presented. However, the last section of the paper would require more experiments (see below for more details) to fully support their conclusions.

Detailed comments:

Section 2.1. Recombinant human cell lines to study in vivo transcription

It would be interesting to know the insertion site of the transgene(s) in the U2OS and HEK293T cells.

We thank the Reviewer for this interesting remark. We did not experimentally determine the insertion sites of the transgenes in our U2OS and HEK293T cell lines, since it was not essential for the goals of our research. Nevertheless, we searched for this information in the literature.

Unfortunately, we could not find any information on the insertion site in the manual of Flp-In™ cell lines (https://assets.thermofisher.com/TFS-Assets/LSG/manuals/flpincell_man.pdf), which were used in the generation of the HEK293T cell line. In this manual, it is mentioned that “The location of the FRT site [the integration site] in each Flp-In™ cell line has not been mapped, but is presumed to have integrated into a transcriptionally active genomic locus (…)”.

We also briefly searched some articles that used cell lines generated using this system (PMID: 37033614, PMID: 37229492), but could not find the insertion sites of the respective transgenes.

In the discussion section of the manuscript, we added that the insertion sites of the transgenes have not been experimentally determined in any of the two cell lines (lines 736-737).

Section 2.2. Human β-globin nascent RNA segregates from its originating genes 

Did the author measure the signal for the gene array in the absence of transcription, before adding the doxycycline? This control would be interesting to know how transcription could modify the structure of the gene array. 

We thank the Reviewer for this insightful observation and are pleased to add this data. We added a new supplementary figure (Figure S3) and its description in the results section (lines 254-270). Figure S3 contains micrographs and 3D reconstructions of the gene array and MS2-labeled nascent RNA in the absence of doxycycline. In this figure, we also provide an analysis of these control results considering the parameters assessed in figures 2-6, S2, and S4-S7, which include the length and intensity of the RNA signals. In contrast with the article by Janicki et al. 2004, we could occasionally observe extensive RNA transcription in the absence of doxycycline.

I understand that the live-cell microscopy was performed 16-24 hours after the transcription was induced. This seems a really long time knowing that previous studies (see for example Janicky et al, Cell. 2004,116(5): 683–698), showed that transcription started as early as 10 minutes after addition of doxycycline. Did the authors performed some kinetics to test how the signal ratio was changing over time? This experiment could be a nice addition to the paper. 

We thank the Reviewer for this interesting observation. Unfortunately, since we have frequently observed visually detectable transcription in the absence of doxycycline (please see Figure S3), we did not find it feasible to perform kinetics assays in our system. As such, we opted to wait at least 16 hours before performing microscopy, to focus on non-kinetic properties related to RNA transcription. We agree that the study by Janicki et al. 2004 is an interesting conceptual basis; however, our system has shown not to be amenable to that approach. We mentioned this limitation in the discussion and cited this article (lines 732-736).

Also, can the authors indicate how long after adding the doxycycline were the cells fixed for the RNA FISH experiment?

We thank the Reviewer for this remark. As recommended, we added this information in the materials and methods section (line 905). Cells were fixed approximately 24 hours after adding doxycycline.

When the authors compare the signal for the gene array and the transcript (fig 2 and 3), they consider the size of the signal and measure the longest diameter of the signal. It would be interesting to repeat the quantification using the intensity of the signal as it should reflect better the level of RNA. This comment is valid for all sections of the paper.

We thank the Reviewer for this insightful observation and agree that signal intensity quantification is a more direct estimation of the Pol II and RNA levels. Accordingly, we performed signal intensity quantifications in sections 2.2 to 2.4 and included their analyses in panels B and C of figures S2 and S4-S7. In Figures 2-6, we also replaced the correlation p-value with the correlation R2 (coefficient of determination). Signal intensity quantification was also added to the microscopy data obtained in the absence of doxycycline (Figure S3). The legends of figures S2 and S4-S7 were updated and the new panels have been described in the results section. The signal intensity quantification protocol used has been added to the materials and methods section (lines 947-956). In the discussion, we also provide a brief comparison (with strengths and limitations) of both quantification methods (lines 623-627).

Section 2.3. mRNA transcription is not homogeneous among identical human β-globin genes

Can the authors mention how long after adding the doxycycline were the cells fixed for the Pol II immunofluorescence experiment?

We thank the Reviewer for this remark. As recommended, we added this information in the materials and methods section (lines 891-892). Cells were fixed approximately 24 hours after adding doxycycline.

By using an antibody directed at the phosphorylated CTD, the authors are only considering the elongating polymerases at the time the cells were fixed. It would be interesting to test the total amount of polymerases recruited to the gene array.

We thank the Reviewer for this insightful observation. We agree that the visualization of total polymerases provides a more complete depiction of their interactions with the gene array. Nevertheless, since our aim was the visualization of RNA transcription, we found it sufficient to visualize only the elongating polymerases. Furthermore, the first author of the manuscript has relocated himself to another research institute and is unable to perform the suggested experiment. Nevertheless, in the part of the discussion section dedicated to potential future steps, we have mentioned this interesting experiment (lines 739-742).

Section 2.4. Human β-globin nascent RNA segregates from RNA Polymerase II 

It would be interesting to test if the overlap between RNA polymerase II and its transcript is more important shortly after transcription was induced.

We thank the Reviewer for this insightful observation. Unfortunately, since our multiple-copy transgene system exhibited some detectable transcription in the absence of doxycycline (please see Figure S3), we were not able to visualize any dynamic alterations associated with the beginning of transcription, such as any differences in signal overlap. We have mentioned the limitation in the discussion section (lines 732-736).

Section 2.5. dCas9 binding to a human β-globin gene is stable over time

The authors conclude this section by saying: “The stable binding of dCas9-GFP to exon 2 of the β-globin gene under high levels of  Pol II transcription (video 2; Figure 7B) suggests that dCas9 might be able to block mRNA transcriptional elongation”  but they do not  know at this point if the RNA polymerase is blocked or if it can pass through the obstacle. I think this sentence should be modified.

We thank the Reviewer for this observation. We modified the sentence to: “Overall, we observed that, under high levels of Pol II transcription, the binding of dCas9-GFP to exon 2 of the β-globin gene is stable over time” (lines 489-490).

Section 2.6. RNA polymerase II transcribes in the presence of DNA-bound dCas9 

In this section, the authors perform Pol II ChIP using an antibody directed against Pol II CTD phosphorylated at ser5. This modification is mostly found in polymerase at the beginning of the elongation (when the first few hundreds NT are transcribed, and is progressively replaced by ser2 phosphorylation. Knowing this, it is not surprising to find RNA Pol II only in promoter and exon I and not in exon 2 and 3. To be really conclusive, this experiment should be repeated using an antibody recognising Pol II CTD phosphorylated at ser2 or ser2 and ser5.

We thank the reviewer for this observation and agree that the experimental design was not the most appropriate to assess the distribution of transcribing Pol II. We also agree that performing ChIP assays using an antibody recognizing Pol II phosphorylated at Ser2 or Ser2+Ser5 would better assess Pol II transcriptional activity. Unfortunately, we are unable to perform this experiment, since the first author of the manuscript has relocated himself to another research institute. Nevertheless, we adapted the underlying hypothesis and biological interpretation of our ChIP results, to keep their conclusion within the reach of its underlying results.

Reframing: we used ChIP to test whether dCas9 binding increases the accumulation of transcriptionally paused Pol II (phosphorylated at Ser5) in the promoter/exon 1 region of the recombinant gene (lines 513-515). Any increase would indicate transcriptional blocking. We compared the levels of paused Pol II between the telomeres negative control and the other conditions. When the gRNA was bound to the template strand of the gene, there was no increase in paused Pol II in the promoter region. However, when the gRNA was switched to the coding strand of exon 1, we observed an increase in paused Pol II levels in the promoter region, indicating a partial blocking of transcription in a stand-specific way of gRNA hybridization. Statistically significant differences have been indicated in Figure 8. Figures 8-10 have been reorganized into Figure 8 (containing all ChIP data) and Figure 9 (containing all ddRT-PCR data). Their legends have been updated. The reframing of the ChIP results has been included (lines 554-536).

We hope that this reframing of the ChIP component will safeguard the validity of this experiment. Otherwise, we will be willing to completely remove the ChIP data. Nevertheless, the maintenance of Pol II transcriptional activity in the presence of dCas9 is already supported by ddRT-PCR results. As such, we are confident that this conclusion will not be compromised by any remaining issues concerning the ChIP results.

For the sake of clarity, we also separated the assessment of paused Pol II accumulation at the promoter region (ChIP) from the maintenance of transcriptional activity (ddRT-PCR). Section 2.6: “RNA polymerase II transcribes in the presence of DNA-bound dCas9” has been divided into two sections: section 2.6: “dCas9 has little effect on the levels of paused RNA polymerase II in the promoter region” (dedicated only to ChIP) and section 2.7: “RNA polymerase II transcribes in the presence of DNA-bound dCas9” (dedicated only to ddRT-PCR).

Also, could the authors indicate the average size of the chromatin fragments after sonication? 

We thank the Reviewer for this observation and are pleased to add this information. We included a new supplementary figure (Figure S8), in which the average size of the chromatin fragments is revealed in a 2% agarose gel. While samples that were not sonicated show a smear spanning a wide kilobase pair range, sonicated samples, although also showing a smear, also have a broad, yet clearly defined band around 200 bp. This new figure is mentioned in line (543) and its protocol in lines 998-1001.

In the discussion, the authors mention that they can observe some transcription in the absence of doxycycline. Could they give more details on the level of transcription observed before induction? Could this affect the interpretation of their last experiment and explain why they observe some transcripts in the presence of dCas9?

We thank the Reviewer for this remark. In the newly added Figure S3, we show the levels of RNA transcription in the absence of doxycycline and provide their quantitative analysis, as described in lines 257-270. Concerning the interpretation of our last experiment, we highlighted that there was some blocking of Pol II by dCas9 upon gRNA hybridization with the coding strand of the gene (lines 559-563).

Reviewer 2 Report

Comments and Suggestions for Authors

In this study, Pesso and Carvalho created cell lines harboring HBB transgenes to study RNAP and nascent transcript compartmentalization and whether tethering dCas9-GFP would impede RNAP elongation. Using the multi-copy cell line, they identified distinct patterns of nascent transcript overlapping with the array indicating that all copies of HBB are not uniformly transcribed. This observation was independently confirmed using FISH to directly visualize the nascent transcript and ChIP to detect RNAP Ser2 phosphorylation. By tethering dCas9-GFP to various regions of the single copy cell line, they found that it did not impede RNAP function.

Overall, this was a thorough and well-written study that furthers understanding of fundamental properties of RNAP function. I have only a few comments that should be addressed prior to publication.

1.     In the dCas9-GFP experiment outlined in Figure 8, it is unclear why the RNAP Ser5 phosphorylated form was interrogated as it is found primarily at the promoter/ first exon region of genes. Although the nascent transcript data in Figure 9 supports the overall conclusion that dCas9-GFP does not impede RNAP elongation, it would be better to test by ChIP whether occupancy of RNAP Ser2 phosphorylated form is affected at exon 2 and 3 +/- dCas9-GFP.

2.     The authors should test whether simultaneously targeting exons 1, 2 and 3 with dCas9-GFP impedes RNAP elongation.

Author Response

In this study, Pessoa and Carvalho created cell lines harboring HBB transgenes to study RNAP and nascent transcript compartmentalization and whether tethering dCas9-GFP would impede RNAP elongation. Using the multi-copy cell line, they identified distinct patterns of nascent transcript overlapping with the array indicating that all copies of HBB are not uniformly transcribed. This observation was independently confirmed using FISH to directly visualize the nascent transcript and ChIP to detect RNAP Ser2 phosphorylation. By tethering dCas9-GFP to various regions of the single copy cell line, they found that it did not impede RNAP function.

Overall, this was a thorough and well-written study that furthers understanding of fundamental properties of RNAP function. I have only a few comments that should be addressed prior to publication.

  1. In the dCas9-GFP experiment outlined in Figure 8, it is unclear why the RNAP Ser5 phosphorylated form was interrogated as it is found primarily at the promoter/ first exon region of genes. Although the nascent transcript data in Figure 9 supports the overall conclusion that dCas9-GFP does not impede RNAP elongation, it would be better to test by ChIP whether occupancy of RNAP Ser2 phosphorylated form is affected at exon 2 and 3 +/- dCas9-GFP.

We thank the reviewer for this observation and agree that the experimental design was not the most appropriate to assess the distribution of transcribing Pol II. We also agree that performing ChIP assays using an antibody recognizing Pol II phosphorylated at Ser2 or Ser2+Ser5 would better assess Pol II transcriptional activity. Unfortunately, we are unable to perform this experiment, since the first author of the manuscript has relocated himself to another research institute. Nevertheless, we adapted the underlying hypothesis and biological interpretation of our ChIP results, to keep their conclusion within the reach of its underlying results.

Reframing: we used ChIP to test whether dCas9 binding increases the accumulation of transcriptionally paused Pol II (phosphorylated at Ser5) in the promoter/exon 1 region of the recombinant gene (lines 513-515). Any increase would indicate transcriptional blocking. We compared the levels of paused Pol II between the telomeres negative control and the other conditions. When the gRNA was bound to the template strand of the gene, there was no increase in paused Pol II in the promoter region. However, when the gRNA was switched to the coding strand of exon 1, we observed an increase in paused Pol II levels in the promoter region, indicating a partial blocking of transcription in a stand-specific way of gRNA hybridization. Statistically significant differences have been indicated in Figure 8. Figures 8-10 have been reorganized into Figure 8 (containing all ChIP data) and Figure 9 (containing all ddRT-PCR data). Their legends have been updated. The reframing of the ChIP results has been included (lines 554-536).

We hope that this reframing of the ChIP component will safeguard the validity of this experiment. Otherwise, we will be willing to completely remove the ChIP data. Nevertheless, the maintenance of Pol II transcriptional activity in the presence of dCas9 is already supported by ddRT-PCR results. As such, we are confident that this conclusion will not be compromised by any remaining issues concerning the ChIP results.

For the sake of clarity, we also separated the assessment of paused Pol II accumulation at the promoter region (ChIP) from the maintenance of transcriptional activity (ddRT-PCR). Section 2.6: “RNA polymerase II transcribes in the presence of DNA-bound dCas9” has been divided into two sections: section 2.6: “dCas9 has little effect on the levels of paused RNA polymerase II in the promoter region” (dedicated only to ChIP) and section 2.7: “RNA polymerase II transcribes in the presence of DNA-bound dCas9” (dedicated only to ddRT-PCR).

  1. The authors should test whether simultaneously targeting exons 1, 2 and 3 with dCas9-GFP impedes RNAP elongation.

We thank the reviewer for this suggestion. We agree that the simultaneous targeting of the three exons would create a more robust barrier to transcriptional elongation and contribute to better understanding the lack of significant polymerase blocking observed in the present study. Unfortunately, we are unable to perform this experiment, since the first author of the manuscript has relocated himself to another research institute. Nevertheless, in the part of the discussion section dedicated to potential future steps, we have mentioned this interesting test (lines 743-747). We are confident that the lack of this experiment does not compromise the major findings of our study.

Round 2

Reviewer 1 Report

Comments and Suggestions for Authors

I thank the author for answering all my comments. I understand they cannot perform the experiment that I suggested and  I find that the changes they have made satisfy the concerns I had with the original version of the paper. Therefore, after careful review of the new version, I have no further concerns with the manuscript.